# Generalization Analysis for Controllable Learning

**Yi-Fan Zhang** [* 1 2]  **Xiao Zhang** [* 3 4]  **Min-Ling Zhang** [2 5]

## Abstract

Controllability has become a critical issue in trustworthy machine learning, as a controllable learner allows for dynamic model adaptation to task requirements during testing. However, existing research lacks a comprehensive understanding of how to effectively measure and analyze the generalization performance of controllable learning methods. In an attempt to move towards this goal from a generalization perspective, we first establish a unified framework for controllable learning. Then, we develop a novel vector-contraction inequality and derive a tight generalization bound for general controllable learning classes, which is independent of the number of task targets except for logarithmic factors and represents the current best-in-class theoretical result. Furthermore, we derive generalization bounds for two typical controllable learning methods: embedding-based and hypernetwork-based methods. We also upper bound the Rademacher complexities of commonly used control and prediction functions, which serve as modular theoretical components for deriving generalization bounds for specific controllable learning methods in practical applications such as recommender systems. Our theoretical results without strong assumptions provide general theoretical guarantees for controllable learning methods and offer new insights into understanding controllability in machine learning.

---

[*]Equal contribution [1]School of Cyber Science and Engineering, Southeast University, Nanjing 210096, China [2]Key Laboratory of Computer Network and Information Integration (Southeast University), Ministry of Education, China [3]Gaoling School of Artificial Intelligence, Renmin University of China, Beijing, China [4]Engineering Research Center of Next-Generation Intelligent Search and Recommendation, MOE [5]School of Computer Science and Engineering, Southeast University, Nanjing 210096, China. Correspondence to: Min-Ling Zhang <zhangml@seu.edu.cn>.

*Proceedings of the 42nd International Conference on Machine Learning*, Vancouver, Canada. PMLR 267, 2025. Copyright 2025 by the author(s).

## 1. Introduction

Controllability has emerged as a crucial aspect of AI (Yampolskiy, 2020; Artificial Intelligence Safety Summit, 2023; A16z, 2025), enabling AI models to be more reliable and adaptable to user needs in dynamic environments. In the context of machine learning, controllability ensures that learners can dynamically adapt to evolving task requirements at test time, a concept known as controllable learning. In many real-world applications, such as information retrieval and recommender systems, the ability to control model behavior at test time is essential for accommodating diverse user preferences, balancing competing objectives, and adapting to dynamic information-seeking needs (Chen et al., 2023; Mysore et al., 2023; Gao et al., 2024; Chang et al., 2023; He et al., 2022). Existing methods for controllable learning including embedding-based methods that adjust model inputs, and hypernetwork-based methods that generate task-specific model parameters, have demonstrated empirical success (Chen et al., 2023; Chang et al., 2023; Shen et al., 2025; Xie et al., 2025).

However, theoretically understanding controllability in machine learning remains an important open question in machine learning theory. Theoretical analysis of controllable learning methods from the perspective of generalization is an important research avenue. A comprehensive generalization analysis of controllable learning faces two main challenges: 1) How to establish the relationship between the generalization bounds and controllability? 2) How to reduce the dependency of the generalization bounds on the number of task targets? *First*, controllable learning often needs to dynamically respond to the task requirement and often involves multiple task targets. How to establish a unified theoretical framework that can formally characterize these factors is the primary issue of theoretical analysis. In addition, the ideal theoretical framework should be general enough to cover existing controllable learning methods and facilitate the development of general analysis methods and theoretical tools for generalization analysis of controllable learning. With an effective unified theoretical framework, we can explicitly introduce controllability in generalization analysis. *Second*, intuitively, the multiple task targets involved in controllable learning suggest that learning will become more difficult as the number of task targets increases, but the empirical success of controllable learning methods suggests that the

impact of increased difficulty should be limited (Chen et al., 2023; He et al., 2022; Chang et al., 2023), meaning that the effective generalization bound is weakly dependent on the number of task targets. It is clear that generalization analysis can promote a better understanding of controllable learning methods.

In this paper, we establish an effective unified theoretical framework and derive tight bounds for controllable learning. Specifically, we develop a novel vector-contraction inequality, which induces the state-of-the-art bound with no dependency on the number of task targets except for logarithmic factors for general controllable learning classes. In addition, we derive general bounds for two typical controllable learning methods, and we also develop modular theoretical results for commonly used control and prediction functions and show that the bounds for the specific controllable learning methods are flexible combinations of these modular theoretical results.

Our goal is to deepen our understanding of controllable learning methods through the systematic generalization analysis. Major contributions of the paper include:

- We establish a unified theoretical framework and develop a novel vector-contraction inequality for controllable learning, which exploits the Lipschitz continuity of losses w.r.t. $\ell_\infty$ norm and can induce tight bounds.

- We derive tight bounds for general controllable learning classes with no dependency on the number of task targets, which provides general theoretical guarantees for controllable learning methods.

- We derive bounds for two typical controllable learning methods and reveal that different manipulation methods based on the input and control function will lead to significant differences in the bounds. The theoretical techniques and modular results on FNN and Transformer here serve as promising theoretical tools for the generalization analysis of controllable learning.

## 2. Related Work

The theoretical analysis in this paper focuses on controllable machine learning methods. Controllable learning emphasizes the model's ability to adapt dynamically at test time based on task requirements to meet specific task targets. It can be broadly categorized into the following two categories.

**Embedding-based controllable learning** focuses on mapping task requirements into embeddings (Gao et al., 2024; Mysore et al., 2023; Penaloza et al., 2024; Chang et al., 2023; Kong et al., 2024), which are then integrated with other inputs into the original model, serving as a pre-processing method during testing. Mysore et al. (2023)

introduced LACE, an embedding-based controllable recommendation model that constructs editable user profiles from human-readable concepts, allowing users to modify them for adaptive recommendations without retraining. Chang et al. (2023) proposed PEPNet to address the imperfectly double seesaw problem by incorporating controllable parameters in dynamic, multi-domain recommendation scenarios. This approach facilitates controllable and personalized predictions of user interactions across diverse tasks and domains.

**Hypernetwork-based controllable learning** generates (partially) new model parameters through a control function based on the given task requirements, replacing the original model parameters to enable adaptive adjustment (Chen et al., 2023; Shen et al., 2023; He et al., 2022; Li et al., 2023; Yan et al., 2022). Since this approach modifies the mapping between task requirements and model parameters during testing, it can also be considered an in-processing method. Chen et al. (2023) proposed a novel framework named CMR, which leverages a feedforward neural network as a hypernetwork to dynamically generate parameters for a re-ranking model based on varying preference weights. This approach enables online adaptability without retraining and has demonstrated positive gains in online A/B tests within e-commerce scenarios. He et al. (2022) presented Hyperprompt, a technique offering controllable task-conditioning of transformers by dynamically adjusting prompts using a hypernetwork for diverse tasks.

Controllable learning has broad applications in information access. Existing theoretical work has focused solely on the approximation properties of controllable learning (Galanti & Wolf, 2020), while the study of its learning properties, particularly generalization analysis, remains an open problem. This paper aims to bridge the gap in the generalization analysis of controllable learning and provide theoretical tools for a broader exploration of its learning properties.

## 3. Preliminaries

Let $[n] := \{1, \ldots, n\}$ for any natural number $n$. In the context of controllable learning, given a dataset $D = \{(\boldsymbol{x}_1, \boldsymbol{y}_1), \ldots, (\boldsymbol{x}_n, \boldsymbol{y}_n)\}$ with $n$ examples which are identically and independently distributed (i.i.d.) from a probability distribution $P$ on $\mathcal{X} \times \mathcal{Y}$, where $\mathcal{X} \subseteq \mathbb{R}^d$ denotes the $d$-dimensional input space and $\mathcal{Y}$ denotes the label space, $\boldsymbol{x} \in \mathcal{X}, \boldsymbol{y} \in \mathcal{Y}$.

### 3.1. Controllable Learning

Unlike traditional learning methods, where the learning objective is fixed during the training phase and cannot change in the testing phase, controllable learning aims to dynamically adapt to newly arrived task requirements during testing, thereby achieving learner controllability. More specifically,

in controllable learning, the learner receives not only the input $\boldsymbol{x} \in \mathcal{X}$ but also the *task requirement* $\boldsymbol{z} \in \mathcal{Z}$ during the testing phase. It adjusts adaptively based on any given task requirement $\boldsymbol{z} \in \mathcal{Z}$, ensuring that the outputs for $c$ task targets are responsive to the dynamic changes in $\boldsymbol{z}$.

Formally, for $\mathcal{Y} \subseteq \mathbb{R}^c$, the task of controllable learning is to solve the problem of multi-output regression with a given task requirement $\boldsymbol{z}$, i.e., the $j$-th *task target* of the learner $\boldsymbol{f}^{\boldsymbol{z}} = (f_1^{\boldsymbol{z}}, \ldots, f_c^{\boldsymbol{z}}) : \mathcal{X} \mapsto \mathbb{R}^c$ corresponds to the regression of $f_j^{\boldsymbol{z}}$, $j \in [c]$. For $\mathcal{Y} \subseteq \{-1, +1\}^c$, each $\boldsymbol{y} = (y_1, \ldots, y_c)$ is a binary vector and $y_j = 1(-1)$ denotes that the $j$-th label is (ir)relevant, $j \in [c]$, and the task of controllable learning is to learn a classifier corresponding to the task requirement $\boldsymbol{z}$, which assigns each instance with a set of relevant labels. A common strategy is to learn a vector-valued function $\boldsymbol{f}^{\boldsymbol{z}} = (f_1^{\boldsymbol{z}}, \ldots, f_c^{\boldsymbol{z}}) : \mathcal{X} \mapsto \mathbb{R}^c$ and derive the classifier by a thresholding function which divides the label space into relevant and irrelevant label sets.

We consider the *prediction function* (also called *controllable learner*) for task target $j \in [c]$ of the general form: $f_j^{\boldsymbol{z}}(\boldsymbol{x}) = \langle \boldsymbol{w}_j, \zeta_j(\phi_j(\boldsymbol{x}, \psi_j(\boldsymbol{z}))) \rangle$, where the nonlinear mapping $\psi_j$ serves as a *control function*, while the *manipulation function* $\phi_j$ represents the nonlinear transformation that integrates the input and control function into the learner, and the nonlinear mapping $\zeta_j$ corresponds to the classifier learned on the controllable representation generated by $\phi_j$. We define a vector-valued function class of controllable learning as:

$$
\begin{aligned}
\mathcal{F} = \{ \boldsymbol{x} \mapsto & \boldsymbol{f}^{\boldsymbol{z}}(\boldsymbol{x}) : \\
& \boldsymbol{f}^{\boldsymbol{z}}(\boldsymbol{x}) = (f_1^{\boldsymbol{z}}(\boldsymbol{x}), \ldots, f_c^{\boldsymbol{z}}(\boldsymbol{x})), \\
& f_j^{\boldsymbol{z}}(\boldsymbol{x}) = \boldsymbol{w}_j^{\top} \zeta_j(\phi_j(\boldsymbol{x}, \psi_j(\boldsymbol{z}))), \\
& \boldsymbol{w} = (\boldsymbol{w}_1, \ldots, \boldsymbol{w}_c) \in \mathbb{R}^{d \times c}, \alpha(\boldsymbol{w}) \leq \Lambda, \\
& \beta(\zeta(\cdot)) \leq A, \gamma(\phi(\cdot, \cdot)) \leq B, \kappa(\psi(\cdot)) \leq C, \\
& \boldsymbol{x} \in \mathcal{X}, \boldsymbol{z} \in \mathcal{Z}, j \in [c], \Lambda, A, B, C > 0 \},
\end{aligned}
\tag{1}
$$

where $\alpha$ represents a functional that constrains weights, $\beta$ represents a functional that constrains nonlinear mappings $\zeta_j$, $\gamma$ represents a functional that constrains nonlinear mappings $\phi_j$, $\kappa$ represents a functional that constrains control functions $\psi_j$. For embedding-based controllable learning methods, the task requirement $\boldsymbol{z}$ can be editable user profiles, the control function $\psi$ often uses Transformers, and the nonlinear mapping $\zeta$ induced by classifier can use FNNs. For hypernetwork-based controllable learning methods, the task requirement $\boldsymbol{z}$ can be a task indicator, the model corresponding to the control function is a hypernetwork, and the nonlinear mapping $\zeta$ induced by classifier can use Transformer-based models.

For any function $\boldsymbol{f}^{\boldsymbol{z}} : \mathcal{X} \mapsto \mathcal{Y}$, the prediction quality on the example $(\boldsymbol{x}, \boldsymbol{y})$ is measured by a loss function $\ell : \mathcal{X} \times \mathcal{Y} \mapsto \mathbb{R}_+$. The goal of controllable learning is to find a hypothesis $\boldsymbol{f}^{\boldsymbol{z}} \in \mathcal{F}$ with good generalization performance from

the dataset $D$ by optimizing the loss $\ell$. The generalization performance is measured by the expected risk: $R(\boldsymbol{f}^{\boldsymbol{z}}) = \mathbb{E}_{(\boldsymbol{x}, \boldsymbol{y}) \sim P}[\ell(\boldsymbol{f}^{\boldsymbol{z}}(\boldsymbol{x}), \boldsymbol{y})]$. We denote the empirical risk w.r.t. training dataset $D$ as $\widehat{R}_D(\boldsymbol{f}^{\boldsymbol{z}}) = \frac{1}{n} \sum_{i=1}^{n} \ell(\boldsymbol{f}^{\boldsymbol{z}}(\boldsymbol{x}_i), \boldsymbol{y}_i)$. In addition, we define the loss function space as $\mathcal{L} = \{\ell(\boldsymbol{f}^{\boldsymbol{z}}(\boldsymbol{x}), \boldsymbol{y}) : \boldsymbol{f}^{\boldsymbol{z}} \in \mathcal{F}\}$. However, the above mentioned loss is typically the 0-1 loss, which is hard to handle in practice. Hence, one usually consider its surrogate losses.

## 3.2. Related Evaluation Metrics

Although controllable learning has been implicitly used in modern information retrieval. However, there is still a lack of specific evaluation metrics to measure the generalization performance of different controllable learning methods. Here we focus on two commonly used evaluation metrics in controllable learning, i.e., weighted Hamming loss and bipartite ranking loss, and define their surrogate losses. The surrogate loss for **weighted Hamming loss** is denoted by:

$$
\ell_W(\boldsymbol{f}^{\boldsymbol{z}}(\boldsymbol{x}), \boldsymbol{y}) = \frac{1}{c} \sum_{j=1}^{c} v_j \ell_j \left( f_j^{\boldsymbol{z}}(\boldsymbol{x}), y_j \right),
$$

where $\ell_j$ is the loss of the $j$-th task target, $v_j$ is the weight for the $j$-th task target and is bounded by $|v_j| \leq V$ for any $j \in [c]$.

For bipartite ranking problems in controllable learning, since they involve pairwise losses, we need to additionally define the corresponding risks. Let $p_j$ be the probability that the samples are relevant to the $j$-th label. $\mathcal{D}_j^+$ denotes the conditional distribution of the samples over $\mathcal{X}$ given that the samples are relevant to the $j$-th label, and $\mathcal{D}_j^-$ denotes the conditional distribution of the samples over $\mathcal{X}$ given that the samples are irrelevant to the $j$-th label. We define the expected risk w.r.t. multi-target bipartite ranking for controllable learning as follows:

$$
R(\boldsymbol{f}^{\boldsymbol{z}}) = \sum_j p_j R(\boldsymbol{f}^{\boldsymbol{z}} | j)
$$

$$
= \sum_j p_j \mathbb{E}_{\substack{\boldsymbol{x}_i \sim \mathcal{D}_j^+ \\ \boldsymbol{x}_i' \sim \mathcal{D}_j^-}} \frac{1}{c} \sum_{j=1}^{c} \ell_j^{0/1} \left( f_j^{\boldsymbol{z}}(\boldsymbol{x}_i) - f_j^{\boldsymbol{z}}(\boldsymbol{x}_i') \right).
$$

In addition, the empirical risk w.r.t. multi-target bipartite ranking is defined as

$$
\widehat{R}_D(\boldsymbol{f}^{\boldsymbol{z}}) = \sum_j \frac{|X_j^+|}{n} \widehat{R}_D(\boldsymbol{f}^{\boldsymbol{z}} | j) =
\tag{2}
$$

$$
\sum_j \frac{|X_j^+|}{n} \frac{1}{|X_j^+||X_j^-|} \sum_{\substack{\boldsymbol{x}_i \in X_j^+ \\ \boldsymbol{x}_i' \in X_j^-}} \frac{1}{c} \sum_{j=1}^{c} \ell_j^{0/1} \left( f_j^{\boldsymbol{z}}(\boldsymbol{x}_i) - f_j^{\boldsymbol{z}}(\boldsymbol{x}_i') \right),
$$

where $X_j^+ = \{\boldsymbol{x}_i \mid y_j = +1, i \in [n]\}$ $(X_j^- = \{\boldsymbol{x}_i' \mid y_j = -1, i \in [n]\})$ corresponds to the set of the samples that are

relevant (irrelevant) to the $j$-th label. The surrogate loss for **multi-target bipartite ranking loss** is denoted by:

$$\ell_R(\boldsymbol{f^z}(\boldsymbol{x}_i, \boldsymbol{x}_i'), \boldsymbol{y}) = \frac{1}{c} \sum_{j=1}^{c} \ell_j \left( f_j^{\boldsymbol{z}}(\boldsymbol{x}_i) - f_j^{\boldsymbol{z}}(\boldsymbol{x}_i') \right),$$

where $\boldsymbol{x}_i$ ($\boldsymbol{x}_i'$) corresponds to the instances that are relevant (irrelevant) to the $j$-th label.

### 3.3. Related Complexity Measures

Here we introduce the related complexity measures involved in our theoretical results. The Rademacher complexity is used to perform generalization analysis for controllable learning.

**Definition 3.1** (Rademacher complexity). Let $\mathcal{F}$ be a class of real-valued functions mapping from $\mathcal{X}$ to $\mathbb{R}$. Let $D = \{\boldsymbol{x}_1, \ldots, \boldsymbol{x}_n\}$ be a set with $n$ i.i.d. samples. The empirical **Rademacher complexity** over $\mathcal{F}$ is defined by

$$\hat{\Re}_D(\mathcal{F}) = \mathbb{E}_{\boldsymbol{\epsilon}} \left[ \sup_{f \in \mathcal{F}} \frac{1}{n} \sum_{i=1}^{n} \epsilon_i f(\boldsymbol{x}_i) \right],$$

where $\epsilon_1, \ldots, \epsilon_n$ are i.i.d. Rademacher random variables. In addition, we define the worst-case Rademacher complexity as $\tilde{\Re}_n(\mathcal{F}) = \sup_{D \in \mathcal{X}^n} \hat{\Re}_D(\mathcal{F})$.

The vector-valued function class $\mathcal{F}$ of controllable learning makes traditional analysis methods developed for the Rademacher complexity of scalar-valued function class invalid. Hence, we need to new analysis tools to convert the Rademacher complexity of a loss function space associated with $\mathcal{F}$ into the Rademacher complexity of a tractable scalar-valued function class. The Rademacher complexity can be bounded by other scale-sensitive complexity measures, e.g. covering number and fat-shattering dimension (Srebro et al., 2010; Zhang & Zhang, 2023). **The relevant definitions are provided in the appendix**.

## 4. General Bounds for Controllable Learning

In this section, we first introduce the assumptions used. Then, we develop a novel vector-contraction inequality for the Rademacher complexity of the vector-valued function class $\mathcal{F}$. Finally, with the novel vector-contraction inequality, we derive bounds for general function classes of controllable learning with no dependency on the number of task targets, up to logarithmic terms, which achieve the state of the art. **The proof sketches and detailed proofs of the theoretical results in this paper are provided in the appendix.**

**Assumption 4.1.** Assume that the input features, the loss function and each task target of the learner are bounded: $\|\boldsymbol{x}_i\|_2 \leq R, \ell(\cdot, \cdot) \leq M, |f_j^{\boldsymbol{z}}(\cdot)| \leq E$ for $i \in [n]$, $j \in [c]$, where $R, M, E > 0$ are constants.

**Assumption 4.2.** Assume that the loss function $\ell$ is $\mu$-Lipschitz continuous w.r.t. the $\ell_\infty$ norm, that is:

$$\left| \ell(\boldsymbol{f}(\boldsymbol{x}), \cdot) - \ell(\boldsymbol{f}'(\boldsymbol{x}), \cdot) \right| \leq \mu \left\| \boldsymbol{f}(\boldsymbol{x}) - \boldsymbol{f}'(\boldsymbol{x}) \right\|_\infty,$$

where $\mu > 0$, $\|\boldsymbol{t}\|_\infty = \max_{j \in [c]} |t_j|$ for $\boldsymbol{t} = (t_1, \ldots, t_c)$.

Assumption 4.1 and 4.2 are mild assumptions. For Assumption 4.1, The boundedness of the input features is easy to satisfy since in practice normalization is often applied to the input features. In addition, for the function class of controllable learning (1), we often use the assumptions $\|\boldsymbol{w}_j\|_2 \leq \Lambda$, $\|\zeta_j(\cdot)\|_2 \leq A$ for any $j \in [c]$ to replace the boundedness of each task target of the learner, i.e., $E := \Lambda A$, and $A$ can be further refined by the specific model used in controllable learning. For Assumption 4.2, the Lipschitz continuity w.r.t. the $\ell_\infty$ norm has been considered in several literature (Foster & Rakhlin, 2019; Lei et al., 2019; Wu et al., 2021; Zhang & Zhang, 2023; 2024a). The following Proposition 4.3 further illustrates that the commonly used loss functions in controllable learning actually satisfy Assumption 4.2.

**Proposition 4.3.** *Assume that the loss of each output task target $\ell_j$ defined in Subsection 3.2 is $\mu$-Lipschitz continuous, then the surrogate weighted Hamming Loss is $\mu V$-Lipschitz w.r.t. the $\ell_\infty$ norm, the surrogate multi-target bipartite ranking loss is $\mu$-Lipschitz w.r.t. the $\ell_\infty$ norm.*

In fact, each task target in controllable learning can be completely different, hence the coupling relationship between the weights of the functions corresponding to each task target needs to be decoupled. To this end, we define a projection operator $p_j : \mathbb{R}^c \mapsto \mathbb{R}$ to project the $c$-dimensional output vector onto the $j$-th coordinate, $p_j \in \mathcal{P}$, $j \in [c]$. Then, we have the projection function class $\mathcal{P}(\mathcal{F}) = \big\{ (j, \boldsymbol{x}) \mapsto p_j(\boldsymbol{f^z}(\boldsymbol{x})) : p_j(\boldsymbol{f^z}(\boldsymbol{x})) = f_j^{\boldsymbol{z}}(\boldsymbol{x}), \boldsymbol{f^z} \in \mathcal{F}, (j, \boldsymbol{x}) \in [c] \times \mathcal{X} \big\}$, which decouples the relationship among different task targets. With Assumption 4.2 and the above definitions, we develop the following novel vector-contraction inequality for controllable learning to show that the Rademacher complexity of the loss function space $\mathcal{L}$ can be bounded by the worst-case Rademacher complexity of the projection function class $\mathcal{P}(\mathcal{F})$:

**Lemma 4.4.** *Let $\mathcal{F}$ be a function class of controllable learning defined by (1). Let Assumptions 4.1 and 4.2 hold. Given a dataset $D$ of size $n$. Then, we have*

$$\hat{\Re}_D(\mathcal{L}) \leq \frac{12M}{\sqrt{n}} + 96\mu\sqrt{c}\tilde{\Re}_{nc}(\mathcal{P}(\mathcal{F})) \times$$

$$\left(1 + \log_2(4en^2c^2\mu^2) \cdot \ln \frac{M\sqrt{n}}{\mu E}\right),$$

*where $\hat{\Re}_D(\mathcal{L}) = \mathbb{E}_{\boldsymbol{\epsilon}} \left[\sup_{\ell \in \mathcal{L}, \boldsymbol{f^z} \in \mathcal{F}} \frac{1}{n} \sum_{i=1}^{n} \epsilon_i \ell(\boldsymbol{f^z}(\boldsymbol{x}_i))\right]$ is the empirical Rademacher complexity of the loss function space associated with $\mathcal{F}$, and $\tilde{\Re}_{nc}(\mathcal{P}(\mathcal{F}))$ is the worst-case Rademacher complexity of the projection function class.*

*Remark* 4.5. The difficulty of theoretical analysis for controllable learning lies in two aspects. First, the development of controllable learning is driven by real-world applications, and the methods developed are quite different. It is difficult to establish a unified theoretical framework to cover all these typical methods. The lack of a unified framework is the most intuitive and primary obstacle to using existing analytical tools to establish an effective theory bound for controllable learning. The definition of the controllable learning function class and the modular theoretical results provided ensure the establishment of a unified theoretical framework. Second, about reducing the dependency on $c$. The analysis of the number of task targets in the bounds can be traced back to a basic bound with a linear dependency on $c$, which comes from the typical vector-contraction lemma in (Maurer, 2016). The dependency on $c$ can be improved to square-root or logarithmic by preserving the coupling among different components, i.e., $\|\boldsymbol{w}\| \leq \Lambda$. However, each task target in controllable learning can be completely different, and the coupling relationship needs to be decoupled. Hence, we introduce the projection operator. We found that the square-root dependency on $c$ is inevitable for $\ell_2$ Lipschitz loss, which essentially comes from the $\sqrt{c}$ factor in the radius of the empirical $\ell_2$ cover of the projection function class, but $\ell_\infty$ Lipschitz continuity of the loss can eliminate it. Hence, the tight bounds with no dependency on $c$, up to logarithmic terms, can be derived.

*Remark* 4.6. The construction of the auxiliary dataset induced by projection functions can reduce the dependency on the output dimension. For the construction of other types of auxiliary datasets under different problem settings and similar ideas, please refer to (Lei et al., 2019; Wu et al., 2021; Mustafa et al., 2021; Hieu et al., 2024; Lei et al., 2023; Graf et al., 2022). Regarding the more desirable constant, the use of discretized variant of Dudley's integral inequality in (Lei et al., 2023) suggests that this can be achieved.

With the vector-contraction inequality above, we can derive the following tight bound for the surrogate weighted Hamming loss:

**Theorem 4.7.** *Let $\mathcal{F}$ be a function class of controllable learning defined by (1). Let Assumptions 4.1 and 4.2 hold. Given a dataset $D$ of size $n$. Then, for the surrogate weighted Hamming loss, for any $0 < \delta < 1$, with probability at least $1 - \delta$, the following holds for any $\boldsymbol{f}^{\boldsymbol{z}} \in \mathcal{F}$:*

$$R(\boldsymbol{f}^{\boldsymbol{z}}) \leq \widehat{R}_D(\boldsymbol{f}^{\boldsymbol{z}}) + 3M\sqrt{\frac{\log \frac{2}{\delta}}{2n}} + \frac{24M}{\sqrt{n}} +$$
$$\frac{192\mu V \Lambda A(1 + \log_2(4en^2c^2\mu^2) \cdot \ln \frac{M\sqrt{n}}{\mu E})}{\sqrt{n}}.$$

*Remark* 4.8. Although Lemma 4.4 contains a $\sqrt{c}$ factor, the term $\widetilde{\Re}_{nc}(\mathcal{P}(\mathcal{F})) \leq \frac{\Lambda A}{\sqrt{nc}}$, which makes the Rademacher

complexity of the loss function space $\mathcal{L}$ actually independent on $c$, and results in a tight $\widetilde{O}(1/\sqrt{n})$ bound with no dependency on the number of task targets. The projection function class combined with the $\ell_\infty$ norm Lipschitz continuity of loss functions is the key to inducing a generalization bound with no dependent on $c$ except for logarithmic factors. Theorem 4.7 shows that when the number of task targets increases, the generalization bound will be slightly larger, i.e., only a logarithmic increase. This means that although the increase in the number of task targets will affect the difficulty of learning, since Theorem 4.7 implies that the increase in difficulty is logarithmic, the real impact may only come from a few important task targets. Therefore, a controllable learning method can eventually obtain a learner with good generalization performance as long as it can learn these task targets well. Our theoretical results can provide general theoretical guarantees for controllable learning methods that can handle many task targets well in practice (Chen et al., 2023; He et al., 2022; Chang et al., 2023).

Since the surrogate multi-target bipartite ranking loss involves pairwise functions, a sequence of pairs of i.i.d. individual observation in (2) is no longer independent, which makes standard techniques in the i.i.d case for traditional Rademacher complexity inapplicable. Inspired by (Clémençon et al., 2008), we convert the non-sum-of-i.i.d pairwise function to a sum-of-i.i.d form by using permutations in U-process. Hence, for each task target, we define the construction method of the set of i.i.d disjoint positive and negative sample pairs for the $j$-th class label as follows:

1. We denote $\left|X_j^+\right|$ and $\left|X_j^-\right|$ as $t_j$ and $u_j$ and $t_j + u_j = n$ for any $j \in [c]$. We denote the number of disjoint positive and negative sample pairs as $s_j = \min\{t_j, u_j\}$.

2. We uniformly select a positive sample from the set of the samples that are relevant to the $j$-th label and select a negative sample from the set of the samples that are irrelevant to the $j$-th label, then construct the selected positive and negative samples into a pair $(\boldsymbol{x}_i^{j+}, \boldsymbol{x}_i^{j-})$.

3. We construct the set of positive and negative sample pairs by matching the samples from the set of the samples that are relevant to the $j$-th label with the samples from the set of the samples that are irrelevant to the $j$-th label until one of the sets of the positive samples and the negative samples exhausts its available samples for selection. We denote the set of i.i.d disjoint positive and negative sample pairs for the $j$-th class label as $D_j$, and $|D_j| = s_j$.

With these definitions, we then derive the tight bound for the surrogate multi-target bipartite ranking loss as follows:

**Theorem 4.9.** *Let $\mathcal{F}$ be a function class of controllable learning defined by (1). Let Assumptions 4.1 and 4.2 hold.*

*Given a dataset $D$ of size $n$. Then, for the surrogate multi-target bipartite ranking loss, for any $0 < \delta < 1$, with probability at least $1 - \delta$, the following holds for any $\boldsymbol{f}^{\boldsymbol{z}} \in \mathcal{F}$:*

$$R(\boldsymbol{f}^{\boldsymbol{z}}) \leq \widehat{R}_D(\boldsymbol{f}^{\boldsymbol{z}}) + \frac{24\sqrt{2}Mq}{\sqrt{s_0}} + 22Mq\sqrt{\frac{\ln\frac{8c}{\delta}}{s_0}} +$$

$$\frac{384\sqrt{2}q\mu\Lambda A \times (1 + \log_2(4es_0^2c^2\mu^2) \cdot \ln\frac{M\sqrt{s_0}}{\mu E})}{\sqrt{s_0}},$$

*where $s_0 = n\min\{\min_j p_j, \min_j(1-p_j)\}$ and $q = \sum_j p_j$.*

*Remark* 4.10. The above bound shows a logarithmic dependency on the number of task targets with a faster convergence rate $\widetilde{O}(1/\sqrt{s_0})$. When the examples of positive and negative classes are balanced, which is the ideal situation, we have $t_j = u_j$ for any $j \in [c]$, $s_0 = \frac{n}{2}$, then the order of the bound is $\widetilde{O}(1/\sqrt{n})$. When class-imbalance occurs, it is obvious that $s_0$ will be smaller than $\frac{n}{2}$. If class-imbalance is more serious, $s_0$ will be smaller, which will lead to a looser bound for the surrogate multi-target ranking loss. It means that when class-imbalance becomes more and more serious, if the learned classifier cannot handle the problem of class-imbalance well, then its performance on the multi-target bipartite ranking loss will be worse.

*Remark* 4.11. The term $\sum_j p_j$ is involved in the above bound, which indicates that each sample is associated with multiple labels, and each task target can correspond to a label in controllable learning. Real-world scenarios and practical tasks imply that the presence of the term $\sum_j p_j$ introduces two challenges. First, in label-sparse tasks, ensuring the applicability of the above theoretical result requires establishing a connection between the term $\sum_j p_j$ and appropriate sparsity conditions, thereby revealing how sparsity influences generalization bounds. Second, in recommendation systems, while users may simultaneously prefer multiple product categories, excessive probability summation could lead to excessively long recommendation lists, thereby diminishing personalization efficacy. Therefore, designing suitable regularization terms may be necessary to address this issue. In future work, we will further explore solutions to the limitations induced by the term $\sum_j p_j$ from these two perspectives.

# 5. General Bounds for Typical Controllable Learning Methods

In this section, we analyze the generalization bounds for two typical controllable learning methods, i.e., embedding-based and hypernetwork-based controllable learning methods. In order to improve the readability and enhance the applicability of the developed theoretical tools, we decompose the process of generalization analysis of these controllable learning methods into multiple modules. In this way, the bounds

corresponding to the specific controllable learning methods are flexible combinations of these modular theoretical components.

Specifically, in Subsection 5.1, we establish refined formal definitions for embedding-based and hypernetwork-based methods (i.e., $\phi_j$ in class (1)) and give their general theoretical analysis methods. In Subsection 5.2, we give the formal definitions of commonly used control and prediction functions (i.e., $\psi_j$ and $\zeta_j$ in class (1)), and derive their Rademacher complexity. In Subsection 5.3, we derive the corresponding generalization bounds for specific controllable learning methods. This process shows how to flexibly apply the modular theoretical results developed in this paper. We show that different manipulation methods based on the input and control function will lead to significant differences in the constant $A$ of the generalization bound in Theorem 4.7 and 4.9.

## 5.1. Theoretical Analysis Methods for Embedding-based and Hypernetwork-based Methods

**Embedding-based Methods**

Embedding-based methods aim to map the task requirement $\boldsymbol{z}$ into a latent space of inputs $\boldsymbol{x}$. This can be seen as a pre-processing technique at test time, where controllable learners adapt to the task requirement solely by modifying the input features, making them easy to implement in tasks like recommender systems or classification.

Formally, for each task target, we denote the output of the control function as $\boldsymbol{a}^j := \psi_j(\boldsymbol{z}) \in \mathbb{R}^{d'}$, $\phi_j$ is a concatenate function denoted as $\text{Concate}(\cdot, \cdot)$, which concatenate $\boldsymbol{x}$ and $\boldsymbol{a}^j$, i.e., $\text{Concate}(\boldsymbol{x}, \boldsymbol{a}^j) = [x_1, \ldots, x_d, a_1^j, \ldots, a_{d'}^j]^\top \in \mathbb{R}^{d+d'}$. Then, a family of $c$ classifiers $f_j$ with $\rho$-Lipschitz nonlinear mapping (i.e., $\zeta_j$ is $\rho$-Lipschitz) are learned on the generated concatenate representations.

With the above definitions, we can derive the upper bound of the worst-case Rademacher complexity for embedding-based controllable learning methods:

**Theorem 5.1.** *Let $\mathcal{F}$ be a function class of embedding-based controllable learning methods defined by (1). Let Assumptions 4.1 and 4.2 hold. Given a dataset $D$ of size $n$. Then, the worst-case Rademacher complexity of the corresponding projection function class can be bounded as: $\widetilde{\mathfrak{R}}_{nc}(\mathcal{P}(\mathcal{F})) \leq \frac{2\rho\Lambda(R+C)}{\sqrt{nc}}.$*

The constant $A$ of the generalization bound in Theorem 4.7 and 4.9 corresponds to $2\rho(R + C)$ here.

**Hypernetwork-based Methods**

Hypernetwork-based methods use a network as the control function $\psi$ to generate the model parameters of the prediction function, effectively allowing the control function to

control and adjust the parameters of the prediction function. These methods can be seen as an in-processing technique at test time, offering flexibility in adapting model parameters to various task requirements. This makes hypernetworks an appealing approach for applications that require dynamic and task-specific parameter adjustments.

Formally, for each task target, the weights of the nonlinear function $\phi_j$ are generated by a hypernetwork $\psi_j$ (i.e., the model corresponding to the control function is a hypernetwork), we denote the generated vector as $\boldsymbol{w}_{\psi_j}$, and $\phi_j$ can be denoted as $\phi_j(\boldsymbol{x}; \boldsymbol{w}_{\psi_j})$. Furthermore, $\boldsymbol{w}_{\psi_j}$ is determined by minimizing the $\mu_{\text{ctrl}}$-Lipschitz loss $\ell_{\text{ctrl}}$ defined on the control function $\psi_j$, and we denoted the loss function space corresponding to $\ell_{\text{ctrl}}$ as $\mathcal{L}_{\text{ctrl}}$. We also define the class of control function $\psi_j$ as $\mathcal{G}_j$, $j \in [c]$, and use $\mathcal{G}^{\otimes c}$ to refer the $c$-fold Cartesian product of the $c$ function classes $\mathcal{G}_j$.

With the above definitions, we can derive the upper bound of the worst-case Rademacher complexity for hypernetwork-based controllable learning methods:

**Theorem 5.2.** *Let $\mathcal{F}$ be a function class of hypernetwork-based controllable learning methods defined by (1). Let Assumptions 4.1 and 4.2 hold. Given a dataset $D$ of size $n$. Then, the worst-case Rademacher complexity of the corresponding projection function class can be bounded as: $\Re_{nc}(\mathcal{P}(\mathcal{H})) \leq \frac{2\rho\Lambda B}{\sqrt{nc}} + \frac{2\mu_{\text{ctrl}}C}{\sqrt{nc}}$, where $\mathcal{H} = \mathcal{F} + \mathcal{L}_{\text{ctrl}} \circ \mathcal{G}^{\otimes c}$ is the whole function class corresponding to the hypernetwork-based methods.*

*Remark* 5.3. Since hypernetwork-based controllable learning method is two-stage, the hypernetwork is used to generate a vector of weights (i.e., model parameters) for each task target in the first stage, and then these generated parameter vectors are used in the learner of the second stage. Therefore, in order to fully consider the capacity of the hypernetwork corresponding to the first stage, it is necessary and appropriate to define the whole function class as the sum of the function classes $\mathcal{H} = \mathcal{F} + \mathcal{L}_{\text{ctrl}} \circ \mathcal{G}^{\otimes c}$ corresponding to the models of these two stages. The bound of the hypernetwork-based method can be obtained by combining Theorem 5.2 with Lemma 4.4, but $\widetilde{\Re}_{nc}(\mathcal{P}(\mathcal{F}))$ in Lemma 4.4 should be replaced by $\widetilde{\Re}_{nc}(\mathcal{P}(\mathcal{H}))$. The constant $A$ of the bound in Theorem 4.7 and 4.9 corresponds to $2\rho B$ here. However, unlike the embedding-based methods, the introduction of the hypernetwork class $\mathcal{G}_j$ leads to an additional increase in complexity, i.e., the last term in Theorem 5.2.

## 5.2. The Rademacher Complexities for Commonly-used Control and Prediction Functions

In this Subsection, we first formally introduce the commonly used control or prediction functions in controllable learning, mainly including two models, i.e., Feedforward Neural Network (FNN) and Transformer, and then derive the upper

bounds of their Rademacher complexities. These theoretical results can be used as components to derive the generalization bounds for specific controllable learning methods.

### Feedforward Neural Network (FNN)

We define a feedforward neural network as follows:

$$f(\boldsymbol{x}) = \boldsymbol{w}^\top \sigma(W_L \sigma(W_{L-1} \cdots \sigma(W_1 \boldsymbol{x}))),$$

where $W_l$ are the parameter metrices, $l \in [L]$, and $\sigma$ is the ReLU activation. With the above definitions, we can derive the upper bound of the Rademacher complexity for FNN:

**Theorem 5.4.** *Let $\mathcal{F}$ be a function class of FNN. Assume that the parameter matrices in FNN are bounded, i.e., $\|W_l\| \leq B_F$ for any $l \in [L]$, where $B_F > 0$ is a constant. Given a dataset $D$ of size $n$. Then, the Rademacher complexity of 5 layer FNN can be bounded as: $\hat{\Re}_D(\mathcal{F}) \leq \frac{2^5 \Lambda B_F^5 R}{\sqrt{n}}$.*

*Remark* 5.5. According to the proof process, it is obvious that $\hat{\Re}_D(\mathcal{F}) \leq 2^L \Lambda B_F^L R / \sqrt{n}$ for $L$ layer FNN. The capacity-based generalization analysis of deep models involved in Subsection 5.2 mainly uses the "peeling" argument, which is a commonly used method in capacity-based theoretical analysis. The main idea of "peeling" is to reduce the complexity bound for $l$ layer networks to a complexity bound for $l-1$ layer networks, and for each reduction, a product factor of a Lipschitz constant for the activation function and an upper bound for the weight matrix norm will be introduced. After applying $l$ reductions, the multiplication of product factors with exponential dependency on the depth may make the bound vacuous (Neyshabur et al., 2015; Bartlett et al., 2017; 2019). How to develop non-vacuous capacity-based generalization analysis methods for deep models is still an open question (Neyshabur et al., 2018; Golowich et al., 2018; Zhang & Zhang, 2023) and we will further explore it in the future. However, our goal here is to provide modular theoretical results for generalization analysis of controllable learning, and the depth of the models used in controllable learning is often limited or even shallow, so the theoretical results in Subsection 5.2 are valid.

### Transformer

We follow the definition and notation of self-attention and Transformer in (Edelman et al., 2022). Let $W_C \in \mathbb{R}^{k \times d}, W_V \in \mathbb{R}^{d \times k}$, and $W_Q, W_K \in \mathbb{R}^{d \times k}$ be trainable weight matrices. Let $X := [\boldsymbol{x}_1, \boldsymbol{x}_2, \ldots, \boldsymbol{x}_T]^\top \in \mathbb{R}^{T \times d}$ be the input, i.e., a sequence of $T$ $d$-dimensional tokens. Let $\sigma$ be an element-wise $L_\sigma$-Lipschitz activation function with $\sigma(0) = 0$. Then, a Transformer layer is defined as $\sigma\left(\text{RowSoftmax}\left(XW_Q W_K^\top X^\top\right) X W_V\right) W_C$, where RowSoftmax applies softmax on each row of its input. For the convenience of analysis, we represent $W_Q W_K^\top$ with a single matrix $W_{QK} \in \mathbb{R}^{d \times d}$. Here we focus on the scalar output from the Transformer. We construct a special extra input in the length-$T$ sequence with a index [CLS], which is

a fixed or trainable vector $x_{[\text{CLS}]} \in \mathbb{R}^d$. Finally, the output at index [CLS] is a linear function $\boldsymbol{w}^\top \boldsymbol{y}_{[\text{CLS}]}$, for a trainable parameter $\boldsymbol{w} \in \mathbb{R}^d$. This setup for scalar output is used in BERT (Devlin et al., 2019) and all of its derivatives. The scalar one layer Transformer is $\boldsymbol{w}^\top \boldsymbol{y}_{[\text{CLS}]}$, where $\boldsymbol{y}_{[\text{CLS}]} = W_C^\top \sigma\left(W_V^\top X^\top \text{softmax}\left(X W_{QK}^\top \boldsymbol{x}_{[\text{CLS}]}\right)\right)$. Then, we have the bound of the Rademacher complexity for Transformer:

**Theorem 5.6.** *Let $\mathcal{F}$ be a function class of Transformer. Assume that the parameter matrices in Transformer are bounded, i.e., $\|\boldsymbol{w}\| \le B_{\boldsymbol{w}}$, $\|W_C\| \le B_{W_C}$, $\|W_V\| \le B_{W_V}$, $\|W_{QK}\| \le B_{W_{QK}}$ and $\|\boldsymbol{x}_t\| \le R$ for any $t \in T$. Given a dataset $D$ of size $n$. Then, the Rademacher complexity of the scalar one layer Transformer can be bounded as: $\hat{\mathfrak{R}}_D(\mathcal{F}) \le \frac{4 B_{\boldsymbol{w}} B_{W_C} B_{W_V} B_{W_{QK}} L_\sigma R^3}{\sqrt{n}}$.*

*Remark* 5.7. Theorem 5.6 implies that the order of the bound for Transformer is $O(\frac{1}{\sqrt{n}})$ with no dependency on the sequence length $T$, which is tight w.r.t. the dependency on $T$ and matches the recent theoretical bound for Transformer (Trauger & Tewari, 2024). The key to reducing the dependency on $T$ is to avoid summation in the norm, where the Lipschitz property of $\text{softmax}$ is used, i.e., Corollary A.7 in (Edelman et al., 2022). The symbol $\|\cdot\|$ denotes Frobenius norm. (Edelman et al., 2022) mainly derives bounds for Transformer through covering numbers. It uses the Lipschitz property of functions to convert the covering number of Transformer function class into those of vector-valued linear classes, and uses the upper bounds of covering numbers for vector-valued linear classes to obtain the result for Transformers. The $\|\cdot\|_{2,1}$ norm involved therein originates from the upper bounds of covering numbers for vector-valued linear classes. Specifically, when upper bounding the covering number of the vector-valued linear class, the summation of covering numbers for scalar-valued linear classes corresponding to each component in the vector-valued linear function induces the $\|\cdot\|_{:,1}$ norm in the $\|\cdot\|_{2,1}$ norm. Our result is derived using the standard Rademacher analysis, following the conventional neural network analysis method, which naturally induce Frobenius norm. We do not emphasize the tightness regarding norm constraints here. Our main purpose here is to reduce the dependency on $T$, so we focus on analyzing a single layer Transformer as an example. For the bound of the deep case, similar to other deep models, the peeling argument is also used for analysis. The dependency of induced bound on depth is similar to existing theoretical results (Edelman et al., 2022; Trauger & Tewari, 2024).

## 5.3. Generalization Bounds for Specific Controllable Learning Methods

Recently, the controllability of AI has become a key issue in many applications, particularly in information access applications such as recommender systems (Joachims, 2024; Shen et al., 2024). This subsection will use state-of-the-art

controllable learning methods in recommender systems as examples to concretize the nonlinear mapping $\zeta$ and the control function $\psi$ of controllable learner in (1), and further provide generalization bounds for embedding-based and hypernetwork-based controllable learning, which are flexible combinations of the above modular theoretical results.

**Bounds for Embedding-based Controllable Learning**

In embedding-based controllable recommendation models, the task description $\boldsymbol{z}$ is often expressed in natural language (e.g., editable user profiles). After being processed by a Transformer language model encoder, the obtained embeddings are used for the subsequent recommendation task (i.e., the control function $\psi$ is typically implemented using a Transformer) (Gao et al., 2024; Mysore et al., 2023; Kong et al., 2024). The nonlinear mapping $\zeta$ is often an FNN (Shen et al., 2021; Chang et al., 2023; Penaloza et al., 2024), which receives the embeddings generated by $\phi$ and aligns them with the user's preference patterns. Then, we have:

**Theorem 5.8.** *Let $\mathcal{F}$ be a function class of embedding-based controllable learning methods defined by (1), where the control function is a Transformer and the classifier is induced by a three-layer FNN. Let Assumptions 4.1 and 4.2 hold. Given a dataset $D$ of size $n$. Then, for the surrogate weighted Hamming loss, for any $0 < \delta < 1$, with probability at least $1 - \delta$, the following holds for any $\boldsymbol{f}^{\boldsymbol{z}} \in \mathcal{F}$:*

$$R(\boldsymbol{f}^{\boldsymbol{z}}) \le \widehat{R}_D(\boldsymbol{f}^{\boldsymbol{z}}) + 3M\sqrt{\frac{\log\frac{2}{\delta}}{2n}} + \frac{24M}{\sqrt{n}} +$$
$$(1 + \log_2(4en^2c^2\mu^2) \cdot \ln\frac{M\sqrt{n}}{\mu E}) \times$$
$$\frac{192\mu V \Lambda 2^3 B_F^3 (R + 4 B_{W_C} B_{W_V} B_{W_{QK}} L_\sigma R^3)}{\sqrt{n}},$$

*where $B_F, B_{W_C}, B_{W_V}, B_{W_{QK}}, L_\sigma > 0$ are constants.*

*Proof.* According to Theorem 5.4 and 5.6, $2\rho$ and $C$ in Theorem 5.1 correspond to $2^3 B_F^3$ and $4 B_{W_C} B_{W_V} B_{W_{QK}} L_\sigma R^3$. This is obvious since the processes of upper bounding these Rademacher complexities are similar. Therefore, we only need to find the upper bounds for $\zeta$ and $\phi$ corresponding to the specific models in Theorem 5.1 and replace them. Then, according to Theorem 5.1, $A := 2\rho(R + C)$. Finally, the desired bound can be derived by replacing $A$ in Theorem 4.7 with $2^3 B_F^3 (R + 4 B_{W_C} B_{W_V} B_{W_{QK}} L_\sigma R^3)$. $\square$

**Bounds for Hypernetwork-based Controllable Learning**

In hypernetwork-based controllable recommendation models, the task description $\boldsymbol{z}$ is often represented as a task indicator or a preference weight vector for various metrics. In such cases, a hypernetwork is typically chosen to be an FNN (Chen et al., 2023; Shen et al., 2023; He et al., 2022; Li

et al., 2023; Yan et al., 2022). The hypernetwork takes $z$ as input and outputs model parameters, which replace certain parameters in the main recommendation model (i.e., nonlinear mapping $\zeta$). In this hypernetwork-based paradigm, the method is model-agnostic, meaning the main recommendation model can be any representative architecture, including Transformer-based models for modeling user behavior sequences (Kang & McAuley, 2018; Zhou et al., 2020; Wu et al., 2020; de Souza Pereira Moreira et al., 2021). Then, we have the following bound for hypernetwork-based models:

**Theorem 5.9.** *Let $\mathcal{F}$ be a function class of hypernetwork-based controllable learning methods defined by (1), where the control function is a three-layer FNN and the classifier is induced by a Transformer. Let Assumptions 4.1 and 4.2 hold. Given a dataset $D$ of size $n$. Then, for the surrogate weighted Hamming loss, for any $0 < \delta < 1$, with probability at least $1 - \delta$, the following holds for any $\boldsymbol{f}^{\boldsymbol{z}} \in \mathcal{F}$:*

$$R(\boldsymbol{f}^{\boldsymbol{z}}) \leq \widehat{R}_D(\boldsymbol{f}^{\boldsymbol{z}}) + 3M\sqrt{\frac{\log \frac{2}{\delta}}{2n}} + \frac{24M}{\sqrt{n}} +$$

$$192\mu V (1 + \log_2(4en^2c^2\mu^2) \cdot \ln \frac{M\sqrt{n}}{\mu E}) \times$$

$$\frac{4B_{W_C}B_{W_V}B_{W_{QK}}L_\sigma R^3\Lambda + 2^4\mu_{\text{ctrl}}\Lambda B_F^3 R}{\sqrt{n}},$$

*where $B_F$, $B_{W_C}$, $B_{W_V}$, $B_{W_{QK}}$, $L_\sigma > 0$ are constants.*

*Proof.* The proof idea is the same as Theorem 5.8, according to Theorem 5.6, $2\rho B$ and $C$ in Theorem 5.2 correspond to $4B_{W_C}B_{W_V}B_{W_{QK}}L_\sigma R^3$ and $2^3\Lambda B_F^3 R$. Then, according to Theorem 5.2, $A := 2\rho B$. However, the introduction of the hypernetwork class leads to an additional last term in Theorem 5.2, hence, for hypernetwork-based methods, $\Lambda A$ in Theorem 4.7 should actually be $2\rho\Lambda B + 2\mu_{\text{ctrl}}C$. Finally, the desired bound can be derived by replacing $\Lambda A$ in Theorem 4.7 with $4B_{W_C}B_{W_V}B_{W_{QK}}L_\sigma R^3\Lambda + 2^4\mu_{\text{ctrl}}\Lambda B_F^3 R$. □

*Remark* 5.10. Theorems 5.8 and 5.9 show that the order of the bounds for specific embedding-based and hypernetwork-based methods is $\widetilde{O}(1/\sqrt{n})$. However, their constants are different. The more complex constant term in Theorem 5.8 corresponds to $C$ in Theorem 5.1, while the more complex constant term in Theorem 5.9 corresponds to the first term in Theorem 5.2. They show that to improve the capacity of models, for embedding-based methods, the control function often chooses models with higher complexity, while for hypernetwork-based methods, since the main model is used for learning, the complexity of models selected for the main model is higher than that of the hypernetwork. This can improve the representation ability of models, so it is easier to learn hypotheses with better generalization in the class. These theoretical results are consistent with the actual meth-

ods and can provide an explanation for the models selected in the actual methods cited in Subsection 5.3.

## 6. Discussion

To understand controllable learning more clearly and intuitively, we provide two real-world examples of controllable learning scenarios: 1) A news aggregator adjusts article rankings in real time using user-specified rules (e.g., exclude gaming content today) to promote diverse topics. The control function modifies inputs/parameters to enforce filters while keeping recommendations relevant. 2) A trading algorithm adapts portfolio strategies based on real-time goals (e.g., protect capital during market drops) to minimize losses and maximize risk-adjusted returns. The control function tunes parameters dynamically to balance objectives without retraining. These examples show how our framework enables systems to adapt inputs/parameters at test time to meet evolving task requirements (e.g., content preferences, market conditions) while ensuring generalization guarantees for targets like diversity and risk management.

Controllable learning and multi-label learning are different learning settings, but there are some connections between them. Specifically, from the perspective of the model output, the outputs of both controllable learning and multi-label learning can be expressed as vector-valued outputs. From the perspective of the model itself and the input, they are different. Controllable learning places more emphasis on task requirements, i.e., the learner can adaptively adjust to dynamically respond to the requirements of different task targets, while multi-label learning does not involve task requirements corresponding to different task targets, so controllable learning is not a special case of multi-label learning. However, when the control function $\psi$ in controllable learning is $\emptyset$, controllable learning will degenerate into a specific multi-label learning method, i.e., multi-label learning based on the label-specific representation learning strategy (Zhang & Zhang, 2024b). At this time, the relevant theoretical results can be generalized to multi-label learning scenarios.

## 7. Conclusion

In this paper, we first establish a unified theoretical framework for controllable learning. Then, we propose a novel vector-contraction inequality and derive a tight bound with no dependency on $c$ for general controllable learning classes. In addition, we derive general bounds for two typical controllable learning methods and develop modular theoretical results for commonly used control and prediction functions.

In future work, we will extend the analysis to more controllable learning methods, e.g., controllable generation models, and derive theoretical results for a broader range of models to enrich our modular theoretical tool set.

## Acknowledgements

The authors wish to thank the anonymous reviewers and the area chair for their helpful comments and suggestions, especially the area chair for his dedicated efforts and selfless assistance in clarifying all ambiguities, ensuring the soundness and enhancing the quality of the paper. This work was supported by the National Science Foundation of China (62225602, 62376275).

## Impact Statement

This paper presents work whose goal is to advance the field of Machine Learning. There are many potential societal consequences of our work, none which we feel must be specifically highlighted here.

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

# A. Appendix

## A.1. Appendix Outline

In the appendix, we give the detailed proofs of our theoretical results in the main paper. Our main proofs include:

- The $\ell_\infty$ Lipschitz continuity of the commonly used losses in controllable learning (Proposition 4.3).

- The novel vector-contraction inequality for $\ell_\infty$ Lipschitz surrogate loss (Lemma 4.4).

- The tight bound of the general controllable learning class for the surrogate weighted Hamming loss (Theorem 4.7).

- The tight bound of the general controllable learning class for the surrogate multi-target bipartite ranking loss (Theorem 4.9).

- The upper bound of the Rademacher complexity for embedding-based controllable learning methods (Theorem 5.1).

- The upper bound of the Rademacher complexity for hypernetwork-based controllable learning methods (Theorem 5.2).

- The upper bound of the Rademacher complexity for FNN (Theorem 5.4).

- The upper bound of the Rademacher complexity for Transformer (Theorem 5.6).

## A.2. Preliminaries

### A.2.1. DEFINITIONS OF THE CORRESPONDING COMPLEXITY MEASURES

**Definition A.1** ($\ell_\infty$ norm covering number). Let $\mathcal{F}$ be a class of real-valued functions mapping from $\mathcal{X}$ to $\mathbb{R}$. Let $D = \{\boldsymbol{x}_1, \ldots, \boldsymbol{x}_n\}$ be a set with $n$ i.i.d. samples. For any $\epsilon > 0$, the empirical $\ell_\infty$ norm covering number $\mathcal{N}_\infty(\epsilon, \mathcal{F}, D)$ w.r.t. $D$ is defined as the minimal number $m$ of a collection of vectors $\boldsymbol{v}^1, \ldots, \boldsymbol{v}^m \in \mathbb{R}^n$ such that $\max_{i \in [n]} \left| f(\boldsymbol{x}_i) - \boldsymbol{v}_i^j \right| \le \epsilon$ ($\boldsymbol{v}_i^j$ is the $i$-th component of the vector $\boldsymbol{v}^j$). In this case, we call $\{\boldsymbol{v}^1, \ldots, \boldsymbol{v}^m\}$ an $(\epsilon, \ell_\infty)$-cover of $\mathcal{F}$ with respect to $D$. We also define $\mathcal{N}_\infty(\epsilon, \mathcal{F}, n) = \sup_D \mathcal{N}_\infty(\epsilon, \mathcal{F}, D)$.

**Definition A.2** (Fat-shattering dimension). Let $\mathcal{F}$ be a class of real-valued functions mapping from $\mathcal{X}$ to $\mathbb{R}$. We define the fat-shattering dimension $\mathrm{fat}_\epsilon(\mathcal{F})$ at scale $\epsilon > 0$ as the largest $p \in \mathbb{N}$ such that there exist $p$ points $\boldsymbol{x}_1, \ldots, \boldsymbol{x}_p \in \mathcal{X}$ and witnesses $s_1, \ldots, s_p \in \mathbb{R}$ satisfying: for any $\delta_1, \ldots, \delta_p \in \{-1, +1\}$ there exists $f \in \mathcal{F}$ with

$$\delta_i \left( f(\boldsymbol{x}_i) - s_i \right) \ge \epsilon, \quad \forall i = 1, \ldots, p.$$

### A.2.2. THE BOUND FOR THE LOSS FUNCTION SPACE

According to McDiarmid's inequality (McDiarmid et al., 1989) and the symmetrization technique, it is easy to obtain that for any training dataset $D = \{(\boldsymbol{x}_i, \boldsymbol{y}_i) : i \in [n]\}$, $\ell(\cdot, \cdot) \le M$, with probability at least $1 - \delta$, the following holds:

$$R(\ell) \le \widehat{R}_D(\ell) + 2\hat{\Re}_D(\mathcal{L}) + 3M\sqrt{\frac{\log \frac{2}{\delta}}{2n}}. \tag{3}$$

## A.3. General Bounds for Controllable Learning

### A.3.1. PROOF OF PROPOSITION 4.3

We first prove that the surrogate weighted Hamming loss is $\mu V$-Lipschitz continuous with respect to the $\ell_\infty$ norm.

$$\left| \ell_W(\boldsymbol{f}^{\boldsymbol{z}}(\boldsymbol{x}), \boldsymbol{y}) - \ell_W\left(\boldsymbol{f}^{\boldsymbol{z}'}(\boldsymbol{x}), \boldsymbol{y}\right) \right|$$

$$= \left| \frac{1}{c} \sum_{j=1}^c v_j \ell_j\left(f_j^{\boldsymbol{z}}(\boldsymbol{x}), y_j\right) - \frac{1}{c} \sum_{j=1}^c v_j \ell_j\left(f_j^{\boldsymbol{z}'}(\boldsymbol{x}), y_j\right) \right|$$

$$= \frac{1}{c} \sum_{j=1}^{c} v_j \left| \ell_j \left( f_j^{\boldsymbol{z}}(\boldsymbol{x}), y_j \right) - \ell_j \left( f_j^{\boldsymbol{z}'}(\boldsymbol{x}), y_j \right) \right|$$

$$\leq V \frac{1}{c} \sum_{j=1}^{c} \mu \left| f_j^{\boldsymbol{z}}(\boldsymbol{x}) - f_j^{\boldsymbol{z}'}(\boldsymbol{x}) \right|$$

$$\leq V \frac{1}{c} \mu c \max_{j \in [c]} \left| f_j^{\boldsymbol{z}}(\boldsymbol{x}) - f_j^{\boldsymbol{z}'}(\boldsymbol{x}) \right|$$

$$= \mu V \left\| \boldsymbol{f}^{\boldsymbol{z}}(\boldsymbol{x}) - \boldsymbol{f}^{\boldsymbol{z}'}(\boldsymbol{x}) \right\|_{\infty}.$$

Then, we prove that the surrogate surrogate multi-target bipartite ranking loss is $\mu$-Lipschitz continuous with respect to the $\ell_{\infty}$ norm.

$$\left| \ell_R(\boldsymbol{f}^{\boldsymbol{z}}(\boldsymbol{x}_i, \boldsymbol{x}_i'), \boldsymbol{y}) - \ell_R(\boldsymbol{f}^{\boldsymbol{z}'}(\boldsymbol{x}_i, \boldsymbol{x}_i'), \boldsymbol{y}) \right|$$

$$= \left| \frac{1}{c} \sum_{j=1}^{c} \ell_j \left( f_j^{\boldsymbol{z}}(\boldsymbol{x}_i) - f_j^{\boldsymbol{z}}(\boldsymbol{x}_i') \right) - \frac{1}{c} \sum_{j=1}^{c} \ell_j \left( f_j^{\boldsymbol{z}'}(\boldsymbol{x}_i) - f_j^{\boldsymbol{z}'}(\boldsymbol{x}_i') \right) \right|$$

$$= \frac{1}{c} \sum_{j=1}^{c} \left| \ell_j \left( f_j^{\boldsymbol{z}}(\boldsymbol{x}_i) - f_j^{\boldsymbol{z}}(\boldsymbol{x}_i') \right) - \ell_j \left( f_j^{\boldsymbol{z}'}(\boldsymbol{x}_i) - f_j^{\boldsymbol{z}'}(\boldsymbol{x}_i') \right) \right|$$

$$= \frac{1}{c} \sum_{j=1}^{c} \mu \left| \left( f_j^{\boldsymbol{z}}(\boldsymbol{x}_i) - f_j^{\boldsymbol{z}}(\boldsymbol{x}_i') \right) - \left( f_j^{\boldsymbol{z}'}(\boldsymbol{x}_i) - f_j^{\boldsymbol{z}'}(\boldsymbol{x}_i') \right) \right|$$

$$= \frac{1}{c} \mu c \max_{j \in [c]} \left| \left( f_j^{\boldsymbol{z}}(\boldsymbol{x}_i) - f_j^{\boldsymbol{z}}(\boldsymbol{x}_i') \right) - \left( f_j^{\boldsymbol{z}'}(\boldsymbol{x}_i) - f_j^{\boldsymbol{z}'}(\boldsymbol{x}_i') \right) \right|$$

$$= \mu \left\| \boldsymbol{f}^{\boldsymbol{z}}(\boldsymbol{x}_i, \boldsymbol{x}_i') - \boldsymbol{f}^{\boldsymbol{z}'}(\boldsymbol{x}_i, \boldsymbol{x}_i') \right\|_{\infty}.$$

### A.3.2. PROOF OF LEMMA 4.4

**Proof Sketch**: First, with the refined Dudley's entropy integral inequality (Ledent et al., 2021), the Rademacher complexity of the loss function space $\mathcal{L}$ can be bounded by the empirical $\ell_{\infty}$ norm covering number. Second, according to the Lipschitz continuity w.r.t the $\ell_{\infty}$ norm, the empirical $\ell_{\infty}$ norm covering number of $\mathcal{F}$ can be bounded by that of $\mathcal{P}(\mathcal{F})$. Third, the empirical $\ell_{\infty}$ norm covering number of $\mathcal{P}(\mathcal{F})$ can be bounded by the worst-case Rademacher complexity of $\mathcal{P}(\mathcal{F})$ through the fat-shattering dimension. Hence, the problem is transferred to the estimation of the worst-case Rademacher complexity. Finally, we estimate the lower bound of the worst-case Rademacher complexity of $\mathcal{P}(\mathcal{F})$, and then combined with the above steps, the Rademacher complexity of the loss function space $\mathcal{L}$ associated with $\mathcal{F}$ can be bounded.

We first introduce the following lemmas:

**Lemma A.3** (Khintchine-Kahane inequality (Lust-Piquard & Pisier, 1991)). *Let $\boldsymbol{v}_1, \ldots, \boldsymbol{v}_n \in \mathcal{H}$, where $\mathcal{H}$ is a Hilbert space with $\| \cdot \|$ being the associated p-th norm. Let $\epsilon_1, \ldots, \epsilon_n$ be a sequence of independent Rademacher variables. Then, for any $p \geq 1$ there holds*

$$\min(\sqrt{p-1}, 1) \left[ \sum_{i=1}^{n} \| \boldsymbol{v}_i \|^2 \right]^{\frac{1}{2}} \leq \left[ \mathbb{E}_{\boldsymbol{\epsilon}} \left\| \sum_{i=1}^{n} \epsilon_i \boldsymbol{v}_i \right\|^p \right]^{\frac{1}{p}} \leq \max(\sqrt{p-1}, 1) \left[ \sum_{i=1}^{n} \| \boldsymbol{v}_i \|^2 \right]^{\frac{1}{2}},$$

*and*

$$\mathbb{E}_{\boldsymbol{\epsilon}} \left\| \sum_{i=1}^{n} \epsilon_i \boldsymbol{v}_i \right\| \geq 2^{-\frac{1}{2}} \left[ \sum_{i=1}^{n} \| \boldsymbol{v}_i \|^2 \right]^{\frac{1}{2}}.$$

**Lemma A.4** (Lemma A.2 in (Srebro et al., 2010)). *For any function class $\mathcal{F}$, any $S$ with a finite sample of size $n$ and any $\epsilon > \hat{\mathfrak{R}}_S(\mathcal{F})$, we have that*

$$\text{fat}_\epsilon(\mathcal{F}) \leq \frac{4n \hat{\mathfrak{R}}_S^2(\mathcal{F})}{\epsilon^2}.$$

**Lemma A.5** (Theorem 12.8 in (Anthony & Bartlett, 2009), (Lei et al., 2023)). *If any function in class $\mathcal{F}$ takes values in $[-B, B]$, then for any $S$ with a finite sample of size $n$, any $\epsilon > 0$ with $\mathrm{fat}_\epsilon(\mathcal{F}) < n$, we have*

$$\log \mathcal{N}_\infty (\epsilon, \mathcal{F}, S) \leq 1 + d \log_2 \frac{4eBn}{d\epsilon} \log \frac{4nB^2}{\epsilon^2},$$

*where $d = \mathrm{fat}_{\epsilon/4}(\mathcal{F})$.*

**Lemma A.6** (Refined Dudley's entropy integral inequality (Ledent et al., 2021)). *Let $\mathcal{F}$ be a real-valued function class with $f \leq B$, $f \in \mathcal{F}$, $B > 0$, and assume that $0 \in \mathcal{F}$. Let $S$ be a finite sample of size $n$. For any $2 \leq p \leq \infty$, we have the following relationship between the Rademacher complexity $\hat{\Re}_S(\mathcal{F})$ and the covering number $\mathcal{N}_p(\epsilon, \mathcal{F}, S)$.*

$$\hat{\Re}_S(\mathcal{F}) \leq \inf_{\alpha > 0} \left( 4\alpha + \frac{12}{\sqrt{n}} \int_\alpha^B \sqrt{\log \mathcal{N}_p(\epsilon, \mathcal{F}, S)} d\epsilon \right).$$

**Step 1**: We first derive the relationship between the empirical $\ell_\infty$ norm covering number $\mathcal{N}_\infty(\epsilon, \mathcal{L}, D)$ and the empirical $\ell_\infty$ norm covering number $\mathcal{N}_\infty(\epsilon, \mathcal{P}(\mathcal{F}), [c] \times D)$.

For the dataset $D = \{(\boldsymbol{x}_1, \boldsymbol{y}_1), \ldots, (\boldsymbol{x}_n, \boldsymbol{y}_n)\}$ with $n$ i.i.d. examples:

$$\max_i |\ell(\boldsymbol{f}^{\boldsymbol{z}}(\boldsymbol{x}_i), \boldsymbol{y}_i) - \ell(\boldsymbol{f}^{\boldsymbol{z}'}(\boldsymbol{x}_i), \boldsymbol{y}_i)|$$
$$\leq \mu \max_i \|\boldsymbol{f}^{\boldsymbol{z}}(\boldsymbol{x}_i) - \boldsymbol{f}^{\boldsymbol{z}'}(\boldsymbol{x}_i)\|_\infty \quad \text{(Use Assumption 4.2)}$$
$$\leq \mu \max_i \max_j |f_j^{\boldsymbol{z}}(\boldsymbol{x}_i) - f_j^{\boldsymbol{z}'}(\boldsymbol{x}_i)|$$
$$\leq \mu \max_i \max_j |p_j(\boldsymbol{f}^{\boldsymbol{z}}(\boldsymbol{x}_i)) - p_j(\boldsymbol{f}^{\boldsymbol{z}'}(\boldsymbol{x}_i)|. \quad \text{(The definition of the projection function class } \mathcal{P}(\mathcal{F}))$$

Then, according to the definition of the empirical $\ell_\infty$ covering number, we have that an empirical $\ell_\infty$ cover of $\mathcal{P}(\mathcal{F})$ at radius $\epsilon/\mu$ is also an empirical $\ell_\infty$ cover of the loss function space $\mathcal{L}$ at radius $\epsilon$, and we can conclude that:

$$\mathcal{N}_\infty (\epsilon, \mathcal{L}, D) \leq \mathcal{N}_\infty \left( \frac{\epsilon}{\mu}, \mathcal{P}(\mathcal{F}), [c] \times D \right). \tag{4}$$

**Step 2**: We show that the empirical $\ell_\infty$ norm covering number of $\mathcal{P}(\mathcal{F})$ can be bounded by the fat-shattering dimension, and the fat-shattering dimension can be bounded by the worst-case Rademacher complexity of $\mathcal{P}(\mathcal{F})$.

According to Lemma A.4, for any $\epsilon > 2\hat{\Re}_{[c] \times D}(\mathcal{P}(\mathcal{F}))$, we have

$$\mathrm{fat}_\epsilon(\mathcal{P}(\mathcal{F})) \leq \frac{4nc\hat{\Re}_{[c] \times D}^2(\mathcal{P}(\mathcal{F}))}{\epsilon^2}.$$

Then, combining with Lemma A.5, for any $\epsilon \in (0, 2E]$, we have

$$\log \mathcal{N}_\infty (\epsilon, \mathcal{P}(\mathcal{F}), [c] \times D) \leq 1 + \mathrm{fat}_{\epsilon/4}(\mathcal{P}(\mathcal{F})) \log_2^2 \frac{8eE^2 nc}{\epsilon^2}$$
$$\leq 1 + \frac{64nc\hat{\Re}_{[c] \times D}^2(\mathcal{P}(\mathcal{F}))}{\epsilon^2} \log_2^2 \frac{8eE^2 nc}{\epsilon^2}$$
$$\leq 1 + \frac{64nc\widetilde{\Re}_{nc}^2(\mathcal{P}(\mathcal{F}))}{\epsilon^2} \log_2^2 \frac{8eE^2 nc}{\epsilon^2}. \tag{5}$$

**Step 3**: According to Assumption 4.1 in the main paper, we can obtain the lower bound of the worst-case Rademacher complexity $\widetilde{\Re}_{nc}(\mathcal{P}(\mathcal{F}))$ by the Khintchine-Kahane inequality with $p = 1$:

$$\widetilde{\Re}_{nc}(\mathcal{P}(\mathcal{F}))$$
$$= \sup_{[c] \times D \in [c] \times \mathcal{X}^n} \hat{\Re}_{[c] \times D}(\mathcal{P}(\mathcal{F}))$$

$$\begin{aligned}
&= \sup_{[c] \times D \in [c] \times \mathcal{X}^n} \mathbb{E}_{\boldsymbol{\epsilon}} \left[ \sup_{p_j(\boldsymbol{f}^{\boldsymbol{z}}(\boldsymbol{x}_i)) \in \mathcal{P}(\mathcal{F})} \frac{1}{nc} \sum_{i=1}^{n} \sum_{j=1}^{c} \epsilon_{ij} p_j(\boldsymbol{f}^{\boldsymbol{z}}(\boldsymbol{x}_i)) \right] \\
&= \sup_{[c] \times D \in [c] \times \mathcal{X}^n} \mathbb{E}_{\boldsymbol{\epsilon}} \left[ \sup_{f_j^{\boldsymbol{z}} \in \mathcal{F}_j} \frac{1}{nc} \sum_{i=1}^{n} \sum_{j=1}^{c} \epsilon_{ij} f_j^{\boldsymbol{z}}(\boldsymbol{x}_i) \right] \\
&= \sup_{\|\zeta_j(\phi_j(\boldsymbol{x}_i, \psi_j(\boldsymbol{z})))\|_2 \leq A : i \in [n], j \in [c]} \frac{1}{nc} \mathbb{E}_{\boldsymbol{\epsilon}} \left[ \sup_{\|\boldsymbol{w}_j\|_2 \leq \Lambda} \sum_{i=1}^{n} \sum_{j=1}^{c} \epsilon_{ij} \langle \boldsymbol{w}_j, \zeta_j(\phi_j(\boldsymbol{x}_i, \psi_j(\boldsymbol{z}))) \rangle \right] \\
&= \sup_{\|\zeta_j(\phi_j(\boldsymbol{x}_i, \psi_j(\boldsymbol{z})))\|_2 \leq A : i \in [n], j \in [c]} \frac{\Lambda}{nc} \mathbb{E}_{\boldsymbol{\epsilon}} \| \sum_{i=1}^{n} \sum_{j=1}^{c} \epsilon_{ij} \zeta_j(\phi_j(\boldsymbol{x}_i, \psi_j(\boldsymbol{z}))) \| \\
&\geq \sup_{\|\zeta_j(\phi_j(\boldsymbol{x}_i, \psi_j(\boldsymbol{z})))\|_2 \leq A : i \in [n], j \in [c]} \frac{\Lambda}{nc} \frac{1}{\sqrt{2}} \left[ \sum_{i=1}^{n} \sum_{j=1}^{c} \|\zeta_j(\phi_j(\boldsymbol{x}_i, \psi_j(\boldsymbol{z})))\|^2 \right]^{\frac{1}{2}} . \quad \text{(Use Lemma A.3)}
\end{aligned}$$

Since $\|\zeta_j(\boldsymbol{\phi}_j(\boldsymbol{x}_i, \psi_j(\boldsymbol{z})))\|_2 \leq A$, we set $\sup_{\|\zeta_j(\phi_j(\boldsymbol{x}_i, \psi_j(\boldsymbol{z})))\|_2 \leq A : i \in [n], j \in [c]} \frac{1}{nc} \left[ \sum_{i=1}^{n} \sum_{j=1}^{c} \|\zeta_j(\phi_j(\boldsymbol{x}_i, \psi_j(\boldsymbol{z})))\|^2 \right]^{\frac{1}{2}} = \frac{A}{\sqrt{nc}}$. So,

$$\widetilde{\mathfrak{R}}_{nc}(\mathcal{P}(\mathcal{F})) \geq \frac{\Lambda A}{\sqrt{2nc}} = \frac{E}{\sqrt{2nc}}. \tag{6}$$

**Step 4**: According to Lemma A.6 and combined with the above steps, we have

$$\begin{aligned}
&\hat{\mathfrak{R}}_D(\mathcal{L}) \\
&\leq \inf_{\alpha > 0} \left( 4\alpha + \frac{12}{\sqrt{n}} \int_{\alpha}^{M} \sqrt{\log \mathcal{N}_{\infty}(\epsilon, \mathcal{L}, D)} d\epsilon \right) \\
&\leq \inf_{\alpha > 0} \left( 4\alpha + \frac{12}{\sqrt{n}} \int_{\alpha}^{M} \sqrt{\log \mathcal{N}_{\infty}(\frac{\epsilon}{\mu}, \mathcal{P}(\mathcal{F}), [c] \times D)} d\epsilon \right) \quad \text{(Use inequality (4))} \\
&\leq \inf_{\alpha > 0} \left( 4\alpha + \frac{12}{\sqrt{n}} \int_{\alpha}^{M} \sqrt{1 + \frac{64nc\mu^2 \widetilde{\mathfrak{R}}_{nc}^2(\mathcal{P}(\mathcal{F}))}{\epsilon^2} \log_2^2 \frac{2eE^2 nc\mu^2}{\widetilde{\mathfrak{R}}_{nc}^2(\mathcal{P}(\mathcal{F}))}} d\epsilon \right) \quad \text{(Use inequality (5))} \\
&\leq \inf_{\alpha > 0} \left( 4\alpha + \frac{12}{\sqrt{n}} \int_{\alpha}^{M} \sqrt{1 + \frac{64nc\mu^2 \widetilde{\mathfrak{R}}_{nc}^2(\mathcal{P}(\mathcal{F}))}{\epsilon^2} \log_2^2(4en^2 c^2 \mu^2)} d\epsilon \right) \\
&\leq \inf_{\alpha > 0} \left( 4\alpha + \frac{12M}{\sqrt{n}} + 96\mu\sqrt{c}\widetilde{\mathfrak{R}}_{nc}(\mathcal{P}(\mathcal{F})) \log_2(4en^2 c^2 \mu^2) \int_{\alpha}^{M} \epsilon^{-1} d\epsilon \right) \\
&\leq \frac{12M}{\sqrt{n}} + \inf_{\alpha > 0} \left( 4\alpha + 96\mu\sqrt{c}\widetilde{\mathfrak{R}}_{nc}(\mathcal{P}(\mathcal{F})) \log_2(4en^2 c^2 \mu^2) \cdot \ln \frac{M}{\alpha} \right) \\
&\leq \frac{12M}{\sqrt{n}} + 96\mu\sqrt{c}\widetilde{\mathfrak{R}}_{nc}(\mathcal{P}(\mathcal{F}))(1 + \log_2(4en^2 c^2 \mu^2) \cdot \ln \frac{M}{24\mu\sqrt{c}\widetilde{\mathfrak{R}}_{nc}(\mathcal{P}(\mathcal{F}))}) \\
&\quad \text{(Choose } \alpha = 24\mu\sqrt{c}\widetilde{\mathfrak{R}}_{nc}(\mathcal{P}(\mathcal{F}))) \\
&\leq \frac{12M}{\sqrt{n}} + 96\mu\sqrt{c}\widetilde{\mathfrak{R}}_{nc}(\mathcal{P}(\mathcal{F}))(1 + \log_2(4en^2 c^2 \mu^2) \cdot \ln \frac{M\sqrt{n}}{\mu E}).
\end{aligned}$$

### A.3.3. PROOF OF THEOREM 4.7

**Proof Sketch**: We upper bound the worst-case Rademacher complexity $\tilde{\mathfrak{R}}_{nc}(\mathcal{P}(\mathcal{F}))$, and then combined with Lemma 4.4 and Proposition 4.3 (i.e., substitute $\mu V$ into $\mu$ in Lemma 4.4), the desired bound can be derived.

We upper bound the worst-case Rademacher complexity $\tilde{\mathfrak{R}}_{nc}(\mathcal{P}(\mathcal{F}))$ as the following:

$$
\begin{aligned}
&\tilde{\mathfrak{R}}_{nc}(\mathcal{P}(\mathcal{F})) \\
&= \sup_{[c] \times D \in [c] \times \mathcal{X}^n} \hat{\mathfrak{R}}_{[c] \times D}(\mathcal{P}(\mathcal{F})) \\
&= \sup_{[c] \times D \in [c] \times \mathcal{X}^n} \mathbb{E}_{\boldsymbol{\epsilon}} \left[ \sup_{p_j(\boldsymbol{f^z}(\boldsymbol{x}_i)) \in \mathcal{P}(\mathcal{F})} \frac{1}{nc} \sum_{i=1}^{n} \sum_{j=1}^{c} \epsilon_{ij} p_j(\boldsymbol{f^z}(\boldsymbol{x}_i)) \right] \\
&= \sup_{[c] \times D \in [c] \times \mathcal{X}^n} \mathbb{E}_{\boldsymbol{\epsilon}} \left[ \sup_{f_j^{\boldsymbol{z}} \in \mathcal{F}_j} \frac{1}{nc} \sum_{i=1}^{n} \sum_{j=1}^{c} \epsilon_{ij} f_j^{\boldsymbol{z}}(\boldsymbol{x}_i) \right] \\
&= \sup_{\|\zeta_j(\phi_j(\boldsymbol{x}_i, \psi_j(\boldsymbol{z})))\|_2 \leq A : i \in [n], j \in [c]} \frac{1}{nc} \mathbb{E}_{\boldsymbol{\epsilon}} \left[ \sup_{\|\boldsymbol{w}_j\|_2 \leq \Lambda} \sum_{i=1}^{n} \sum_{j=1}^{c} \epsilon_{ij} \langle \boldsymbol{w}_j, \zeta_j(\phi_j(\boldsymbol{x}_i, \psi_j(\boldsymbol{z}))) \rangle \right] \\
&= \sup_{\|\zeta_j(\phi_j(\boldsymbol{x}_i, \psi_j(\boldsymbol{z})))\|_2 \leq A : i \in [n], j \in [c]} \frac{\Lambda}{nc} \mathbb{E}_{\boldsymbol{\epsilon}} \| \sum_{i=1}^{n} \sum_{j=1}^{c} \epsilon_{ij} \zeta_j(\phi_j(\boldsymbol{x}_i, \psi_j(\boldsymbol{z}))) \| \\
&\leq \sup_{\|\zeta_j(\phi_j(\boldsymbol{x}_i, \psi_j(\boldsymbol{z})))\|_2 \leq A : i \in [n], j \in [c]} \frac{\Lambda}{nc} \left[ \mathbb{E}_{\boldsymbol{\epsilon}} \| \sum_{i=1}^{n} \sum_{j=1}^{c} \epsilon_{ij} \zeta_j(\phi_j(\boldsymbol{x}_i, \psi_j(\boldsymbol{z}))) \|^2 \right]^{\frac{1}{2}} \quad \text{(Use Jensen's Inequality)} \\
&\leq \sup_{\|\zeta_j(\phi_j(\boldsymbol{x}_i, \psi_j(\boldsymbol{z})))\|_2 \leq A : i \in [n], j \in [c]} \frac{\Lambda}{nc} \left[ \sum_{i=1}^{n} \sum_{j=1}^{c} \|\zeta_j(\phi_j(\boldsymbol{x}_i, \psi_j(\boldsymbol{z})))\|^2 \right]^{\frac{1}{2}} \leq \frac{\Lambda A}{\sqrt{nc}}. \quad \text{(Use Lemma A.3)} \quad (7)
\end{aligned}
$$

Then, combining with Proposition 4.3, we have

$$
\begin{aligned}
\hat{\mathfrak{R}}_D(\mathcal{L}) &\leq \frac{12M}{\sqrt{n}} + 96 \mu V \sqrt{c} \tilde{\mathfrak{R}}_{nc}(\mathcal{P}(\mathcal{F}))(1 + \log_2(4en^2c^2\mu^2) \cdot \ln \frac{M\sqrt{n}}{\mu E}) \\
&\leq \frac{12M}{\sqrt{n}} + \frac{96 \mu V \Lambda A (1 + \log_2(4en^2c^2\mu^2) \cdot \ln \frac{M\sqrt{n}}{\mu E})}{\sqrt{n}}.
\end{aligned}
$$

Combining with (3), then

$$
R(\boldsymbol{f^z}) \leq \widehat{R}_D(\boldsymbol{f^z}) + \frac{24M}{\sqrt{n}} + \frac{192 \mu V \Lambda A (1 + \log_2(4en^2c^2\mu^2) \cdot \ln \frac{M\sqrt{n}}{\mu E})}{\sqrt{n}} + 3M \sqrt{\frac{\log \frac{2}{\delta}}{2n}}.
$$

### A.3.4. PROOF OF THEOREM 4.9

**Proof Sketch**: First, for the surrogate multi-target bipartite ranking loss, by using the U-process technique, we define the empirical Rademacher complexity of a loss function space associated with the controllable learning class $\mathcal{F}$ over the set of i.i.d disjoint positive and negative sample pairs for each $j$-th label, then with two-sided multiplicative Chernoff bound, the generalization error can be bounded by $\hat{\mathfrak{R}}_{D_0}(\mathcal{L}) = \mathbb{E}_{\boldsymbol{\epsilon}} \left[ \sup_{\boldsymbol{f^z} \in \mathcal{F}} \frac{1}{s_0} \sum_{i=1}^{s_0} \frac{1}{c} \sum_{j=1}^{c} \epsilon_i \ell_j \left( f_j^{\boldsymbol{z}}(\boldsymbol{x}_i) - f_j^{\boldsymbol{z}}(\boldsymbol{x}_i') \right) \right]$, where $|D_0| = s_0$ is the number of positive and negative sample pairs that can be constructed for any label. Second, combining with Lemma 4.4 and Proposition 4.3, we have $\hat{\mathfrak{R}}_{D_0}(\mathcal{L}) \leq \frac{12M}{\sqrt{s_0}} + 96 \mu \sqrt{c} \tilde{\mathfrak{R}}_{s_0 c}(\mathcal{P}(\mathcal{F})) \times (1 + \log_2(4es_0^2c^2\mu^2) \cdot \ln \frac{M\sqrt{s_0}}{\mu E})$. Finally, we upper bound the worst-case Rademacher complexity $\tilde{\mathfrak{R}}_{s_0 c}(\mathcal{P}(\mathcal{F})) \leq \frac{2\Lambda A}{\sqrt{s_0 c}}$, the desired bound can be derived.

According to the definitions of bipartite ranking problems in controllable learning in Subsection 3.2, for the surrogate multi-target bipartite ranking loss, we have

$$\begin{aligned}
& R(\boldsymbol{f^z}) - \widehat{R}_D(\boldsymbol{f^z}) \\
=& \sum_j p_j R(\boldsymbol{f^z}|j) - \sum_j \frac{t_j}{n} \widehat{R}_D(\boldsymbol{f^z}|j) \\
=& \sum_j p_j R(\boldsymbol{f^z}|j) - \sum_j p_j \widehat{R}_D(\boldsymbol{f^z}|j) + \sum_j p_j \widehat{R}_D(\boldsymbol{f^z}|j) - \sum_j \frac{t_j}{n} \widehat{R}_D(\boldsymbol{f^z}|j) \\
=& \sum_j p_j \left( R(\boldsymbol{f^z}|j) - \widehat{R}_D(\boldsymbol{f^z}|j) \right) + \sum_j (p_j - \sum_j \frac{t_j}{n}) \widehat{R}_D(\boldsymbol{f^z}|j) \\
\leq& \sum_j p_j \left| R(\boldsymbol{f^z}|j) - \widehat{R}_D(\boldsymbol{f^z}|j) \right| + \sum_j |p_j - \frac{t_j}{n}| M
\end{aligned} \tag{8}$$

Rademacher complexity has proved to be a powerful data-dependent measure of hypothesis space complexity. However, since the surrogate multi-target bipartite ranking loss involves pairwise functions, a sequence of pairs of i.i.d. individual observation in (2) is no longer independent, which makes standard techniques in the i.i.d case for traditional Rademacher complexity inapplicable. We convert the non-sum-of-i.i.d pairwise function to a sum-of-i.i.d form by using permutations in U-process (Clémençon et al., 2008).

For each task target, we define the empirical Rademacher complexity of a loss function space associated with the controllable learning class $\mathcal{F}$ over the set of i.i.d disjoint positive and negative sample pairs for the $j$-th label as follows:

$$\hat{\Re}_{D_j}(\mathcal{L}) = \mathbb{E}_{\boldsymbol{\epsilon}|j} \left[ \sup_{\boldsymbol{f^z} \in \mathcal{F}} \frac{1}{s_j} \sum_{i=1}^{s_j} \epsilon_i \ell_R(\boldsymbol{f^z}(\boldsymbol{x}_i^{j+}, \boldsymbol{x}_i^{j-})) \right] = \mathbb{E}_{\boldsymbol{\epsilon}|j} \left[ \sup_{\boldsymbol{f^z} \in \mathcal{F}} \frac{1}{s_j} \sum_{i=1}^{s_j} \frac{1}{c} \sum_{j=1}^{c} \epsilon_i \ell_j \left( f_j^{\boldsymbol{z}}(\boldsymbol{x}_i^{j+}) - f_j^{\boldsymbol{z}}(\boldsymbol{x}_i^{j-}) \right) \right], \quad (9)$$

where each $\epsilon_i$ is an independent Rademacher random variable. The corresponding expected Rademacher complexity is defined as $\Re_{s_j}(\mathcal{L}) = \mathbb{E}_{D|j} \hat{\Re}_{D_j}(\mathcal{L})$.

We then proof the following lemma:

**Lemma A.7.** *Let $q_\tau : \mathcal{X} \times \mathcal{X} \mapsto \mathbb{R}$ be real-valued functions indexed by $\tau \in \mathcal{T}$ where $\mathcal{T}$ is some set. If $\boldsymbol{x}_1, \ldots, \boldsymbol{x}_t$ and $\boldsymbol{x}'_1, \ldots, \boldsymbol{x}'_u$ are i.i.d., $s = \min\{t, u\}$, then for any convex non-decreasing function $\psi$,*

$$\mathbb{E}\psi \left( \sup_{\tau \in \mathcal{T}} \frac{1}{tu} \sum_{i=1}^{t} \sum_{j=1}^{u} q_\tau \left( \boldsymbol{x}_i, \boldsymbol{x}'_j \right) \right) \leq \mathbb{E}\psi \left( \sup_{\tau \in \mathcal{T}} \frac{1}{s} \sum_{i=1}^{s} q_\tau \left( \boldsymbol{x}_i, \boldsymbol{x}'_i \right) \right).$$

*Proof.* The proof of this lemma is inspired by (Clémençon et al., 2008).

$$\begin{aligned}
& \mathbb{E}\psi \left( \sup_{\tau \in \mathcal{T}} \frac{1}{tu} \sum_{i=1}^{t} \sum_{j=1}^{u} q_\tau \left( \boldsymbol{x}_i, \boldsymbol{x}'_j \right) \right) \\
=& \mathbb{E}\psi \left( \sup_{\tau \in \mathcal{T}} \frac{1}{t!} \sum_{\pi_{\boldsymbol{x}}} \frac{1}{u!} \sum_{\pi_{\boldsymbol{x}'}} \frac{1}{s} \sum_{i=1}^{s} q_\tau \left( \boldsymbol{x}_{\pi(i)}, \boldsymbol{x}'_{\pi(i)} \right) \right) \\
\leq& \mathbb{E}\psi \left( \frac{1}{t!} \sum_{\pi_{\boldsymbol{x}}} \frac{1}{u!} \sum_{\pi_{\boldsymbol{x}'}} \sup_{\tau \in \mathcal{T}} \frac{1}{s} \sum_{i=1}^{s} q_\tau \left( \boldsymbol{x}_{\pi(i)}, \boldsymbol{x}'_{\pi(i)} \right) \right) && (\psi \text{ is nondecreasing}) \\
\leq& \frac{1}{t!} \sum_{\pi_{\boldsymbol{x}}} \frac{1}{u!} \sum_{\pi_{\boldsymbol{x}'}} \mathbb{E}\psi \left( \sup_{\tau \in \mathcal{T}} \frac{1}{s} \sum_{i=1}^{s} q_\tau \left( \boldsymbol{x}_{\pi(i)}, \boldsymbol{x}'_{\pi(i)} \right) \right) && (\text{Jensen's inequality}) \\
=& \mathbb{E}\psi \left( \sup_{\tau \in \mathcal{T}} \frac{1}{s} \sum_{i=1}^{s} q_\tau \left( \boldsymbol{x}_i, \boldsymbol{x}'_i \right) \right).
\end{aligned}$$

$\square$

According to Lemma A.7 and the symmetrization technique, we can obtain

$$\mathbb{E}_{D|j}\psi\left(\sup_{\boldsymbol{f^z}\in\mathcal{F}}\left|R(\boldsymbol{f^z}|j) - \widehat{R}_D(\boldsymbol{f^z}|j)\right|\right) \leq \mathbb{E}_{D|j,\boldsymbol{\epsilon}|j}\psi\left(2\sup_{\boldsymbol{f^z}\in\mathcal{F}}\left|\frac{1}{s_j}\sum_{i=1}^{s_j}\epsilon_i\ell_R(\boldsymbol{f^z}(\boldsymbol{x}_i^{j+},\boldsymbol{x}_i^{j-}))\right|\right). \tag{10}$$

We denote $\sup_{\boldsymbol{f^z}\in\mathcal{F}}\left|\frac{1}{s_j}\sum_{i=1}^{s_j}\epsilon_i\ell_R(\boldsymbol{f^z}(\boldsymbol{x}_i^{j+},\boldsymbol{x}_i^{j-}))\right|$ as $H_{\mathcal{L}}^{D_j}$.

The details of inequality (10) are as follows:

$$\mathbb{E}_{D|j}\psi\left(\sup_{\boldsymbol{f^z}\in\mathcal{F}}\left|R(\boldsymbol{f^z}|j) - \widehat{R}_D(\boldsymbol{f^z}|j)\right|\right)$$

$$\leq\mathbb{E}_{D|j}\psi\left(\sup_{\boldsymbol{f^z}\in\mathcal{F}}\left|\frac{1}{|X_j^+||X_j^-|}\sum_{\substack{\boldsymbol{x}_i^{j+}\in X_j^+ \\ \boldsymbol{x}_k^{j-}\in X_j^-}}\left(\ell_R(\boldsymbol{f^z}(\boldsymbol{x}_i^{j+},\boldsymbol{x}_k^{j-})) - R(\boldsymbol{f^z}|j)\right)\right|\right)$$

$$\leq\mathbb{E}_{D|j}\psi\left(\sup_{\boldsymbol{f^z}\in\mathcal{F}}\left|\frac{1}{t_j!\cdot u_j!}\sum_{\pi_{j+},\pi_{j-}}\frac{1}{s_j}\sum_{i=1}^{s_j}\left(\ell_R(\boldsymbol{f^z}(\boldsymbol{x}_{\pi_{j+}(i)}^{j+},\boldsymbol{x}_{\pi_{j-}(i)}^{j-})) - R(\boldsymbol{f^z}|j)\right)\right|\right)$$

$$\leq\mathbb{E}_{D|j}\psi\left(\sup_{\boldsymbol{f^z}\in\mathcal{F}}\frac{1}{t_j!\cdot u_j!}\sum_{\pi_{j+},\pi_{j-}}\left|\frac{1}{s_j}\sum_{i=1}^{s_j}\left(\ell_R(\boldsymbol{f^z}(\boldsymbol{x}_{\pi_{j+}(i)}^{j+},\boldsymbol{x}_{\pi_{j-}(i)}^{j-})) - R(\boldsymbol{f^z}|j)\right)\right|\right)$$

$$\leq\mathbb{E}_{D|j}\psi\left(\frac{1}{t_j!\cdot u_j!}\sum_{\pi_{j+},\pi_{j-}}\sup_{\boldsymbol{f^z}\in\mathcal{F}}\left|\frac{1}{s_j}\sum_{i=1}^{s_j}\left(\ell_R(\boldsymbol{f^z}(\boldsymbol{x}_{\pi_{j+}(i)}^{j+},\boldsymbol{x}_{\pi_{j-}(i)}^{j-})) - R(\boldsymbol{f^z}|j)\right)\right|\right) \quad (\psi \text{ is nondecreasing})$$

$$\leq\frac{1}{t_j!\cdot u_j!}\sum_{\pi_{j+},\pi_{j-}}\mathbb{E}_{D|j}\psi\left(\sup_{\boldsymbol{f^z}\in\mathcal{F}}\left|\frac{1}{s_j}\sum_{i=1}^{s_j}\left(\ell_R(\boldsymbol{f^z}(\boldsymbol{x}_{\pi(i)}^{j+},\boldsymbol{x}_{\pi(i)}^{j-})) - R(\boldsymbol{f^z}|j)\right)\right|\right) \quad (\text{Jensen's inequality})$$

$$\leq\mathbb{E}_{D|j}\psi\left(\sup_{\boldsymbol{f^z}\in\mathcal{F}}\left|\frac{1}{s_j}\sum_{i=1}^{s_j}\left(\ell_R(\boldsymbol{f^z}(\boldsymbol{x}_i^{j+},\boldsymbol{x}_i^{j-})) - R(\boldsymbol{f^z}|j)\right)\right|\right)$$

$$\leq\mathbb{E}_{D|j}\psi\left(\sup_{\boldsymbol{f^z}\in\mathcal{F}}\left|\frac{1}{s_j}\sum_{i=1}^{s_j}\left(\ell_R(\boldsymbol{f^z}(\boldsymbol{x}_i^{j+},\boldsymbol{x}_i^{j-})) - \mathbb{E}_{D|j}\ell_R(\boldsymbol{f^z}(\boldsymbol{x}_i^{j+},\boldsymbol{x}_i^{j-}))\right)\right|\right)$$

$$\leq\mathbb{E}_{D|j}\psi\left(\sup_{\boldsymbol{f^z}\in\mathcal{F}}\left|\frac{1}{s_j}\sum_{i=1}^{s_j}\left(\ell_R(\boldsymbol{f^z}(\boldsymbol{x}_i^{j+},\boldsymbol{x}_i^{j-})) - \mathbb{E}_{D'|j}\ell_R(\boldsymbol{f^z}(\boldsymbol{x}_i^{j+'},\boldsymbol{x}_i^{j-'}))\right)\right|\right)$$

$(D'|j$ is the set of samples with the same distribution as $D|j)$

$$\leq\mathbb{E}_{D|j,D'|j}\psi\left(\sup_{\boldsymbol{f^z}\in\mathcal{F}}\left|\frac{1}{s_j}\sum_{i=1}^{s_j}\left(\ell_R(\boldsymbol{f^z}(\boldsymbol{x}_i^{j+},\boldsymbol{x}_i^{j-})) - \ell_R(\boldsymbol{f^z}(\boldsymbol{x}_i^{j+'},\boldsymbol{x}_i^{j-'}))\right)\right|\right) \quad (\text{Jensen's inequality})$$

$$\leq\mathbb{E}_{D|j,D'|j,\boldsymbol{\epsilon}|j}\psi\left(\sup_{\boldsymbol{f^z}\in\mathcal{F}}\left|\frac{1}{s_j}\sum_{i=1}^{s_j}\epsilon_i\left(\ell_R(\boldsymbol{f^z}(\boldsymbol{x}_i^{j+},\boldsymbol{x}_i^{j-})) - \ell_R(\boldsymbol{f^z}(\boldsymbol{x}_i^{j+'},\boldsymbol{x}_i^{j-'}))\right)\right|\right)$$

$$\leq\frac{1}{2}\mathbb{E}_{D|j,\boldsymbol{\epsilon}|j}\psi\left(2\sup_{\boldsymbol{f^z}\in\mathcal{F}}\left|\frac{1}{s_j}\sum_{i=1}^{s_j}\epsilon_i\ell_R(\boldsymbol{f^z}(\boldsymbol{x}_i^{j+},\boldsymbol{x}_i^{j-}))\right|\right) + \frac{1}{2}\mathbb{E}_{D'|j,\boldsymbol{\epsilon}|j}\psi\left(2\sup_{\boldsymbol{f^z}\in\mathcal{F}}\left|\frac{1}{s_j}\sum_{i=1}^{s_j}-\epsilon_i\ell_R(\boldsymbol{f^z}(\boldsymbol{x}_i^{j+'},\boldsymbol{x}_i^{j-'}))\right|\right)$$

$$\leq\mathbb{E}_{D|j,\boldsymbol{\epsilon}|j}\psi\left(2\sup_{\boldsymbol{f^z}\in\mathcal{F}}\left|\frac{1}{s_j}\sum_{i=1}^{s_j}\epsilon_i\ell_R(\boldsymbol{f^z}(\boldsymbol{x}_i^{j+},\boldsymbol{x}_i^{j-}))\right|\right) \quad (D'|j \text{ with the same distribution as } D|j).$$

First, since the maximum difference caused by replacing one element in $D_j$ or $\epsilon_i|j$ is $\frac{2M}{s_j}$, according to McDiarmid's

inequality, we have

$$P(|H_{\mathcal{L}}^{D_j} - \Re_{s_j}(\mathcal{L})| \geq \varepsilon) \leq 2e^{-\frac{\varepsilon^2 s_j}{2M^2}}.$$

Then, according to the tail bound for sub-Gaussian random variables and Theorem 2.1 in (Boucheron et al., 2013), $H_{\mathcal{L}}^{D_j}$ is a sub-Gaussian random variable with variance proxy $16\frac{M^2}{s_j}$. With the definition of the sub-Gaussian random variable, we have

$$\mathbb{E}_{D|j,\boldsymbol{\epsilon}|j}e^{tH_{\mathcal{L}}^{D_j}} \leq e^{t\Re_{s_j}(\mathcal{L})+\frac{8t^2 M^2}{s_j}}, \ \forall t > 0. \tag{11}$$

Then, for any $\varepsilon > 0$, we have

$$P\left(\sup_{\boldsymbol{f^z}\in\mathcal{F}}\left|R(\boldsymbol{f^z}|j) - \widehat{R}_D(\boldsymbol{f^z}|j)\right| \geq \varepsilon|j\right)$$

$$=P\left(e^{t\sup_{\boldsymbol{f^z}\in\mathcal{F}}|R(\boldsymbol{f^z}|j)-\widehat{R}_D(\boldsymbol{f^z}|j)|} \geq e^{t\varepsilon}|j\right)$$

$$\leq \frac{\mathbb{E}_{D|j}e^{(t\sup_{\boldsymbol{f^z}\in\mathcal{F}}|R(\boldsymbol{f^z}|j)-\widehat{R}_D(\boldsymbol{f^z}|j)|)}}{e^{t\varepsilon}} \quad \text{(Use Markov's Inequality)}$$

$$\leq \frac{\mathbb{E}_{D|j,\boldsymbol{\epsilon}|j}e^{2tH_{\mathcal{L}}^{D_j}}}{e^{t\varepsilon}} \quad \text{(Use inequality (10) with } \psi(x)=e^{tx})$$

$$\leq \frac{e^{2t\Re_{s_j}(\mathcal{L})+\frac{32t^2 M^2}{s_j}}}{e^{t\varepsilon}} \quad \text{(Use inequality (11))}.$$

We set $\frac{e^{2t\Re_{s_j}(\mathcal{L})+\frac{32t^2 M^2}{s_j}}}{e^{t\varepsilon}} = \delta$, then we have

$$\varepsilon = 2\Re_{s_j}(\mathcal{L}) + \frac{32tM^2}{s_j} + \frac{\ln\frac{1}{\delta}}{t}.$$

Hence, we upper bound the term with probability at least $1-\delta$

$$\sup_{\boldsymbol{f^z}\in\mathcal{F}}\left|R(\boldsymbol{f^z}|j) - \widehat{R}_D(\boldsymbol{f^z}|j)\right|$$

$$\leq 2\Re_{s_j}(\mathcal{L}) + \frac{32tM^2}{s_j} + \frac{\ln\frac{1}{\delta}}{t}$$

$$\leq 2\Re_{s_j}(\mathcal{L}) + 8M\sqrt{\frac{2\ln\frac{1}{\delta}}{s_j}}. \tag{12}$$

Next, we upper bound the term $\sum_j |p_j - \frac{t_j}{n}|$.

First, since $t_j \sim \text{Binomial}(n, p_j)$, we have $\mathbb{E} = np_j$. With the two-sided multiplicative Chernoff bound, we have

$$P\left(|t_j - np_j| \geq rnp_j\right) \leq 2e^{-\frac{r^2 np_j}{3}}, \ \forall r \in (0, 1).$$

Then, we have

$$P\left(\sum_j |\frac{t_j}{n} - p_j| \geq r\right)$$

$$\leq P\left(\cup_j \{|\frac{t_j}{n} - p_j| \geq rp_j\}\right)$$

$$\leq \sum_j P\left(|\frac{t_j}{n} - p_j| \geq rp_j\right) \quad \text{(Use Union Bound Inequality)}$$

$$\leq \sum_j 2e^{-\frac{r^2 np_j}{3}}$$

$$\leq 2ce^{-\frac{r^2 n \min_j p_j}{3}}.$$

We set $2ce^{-\frac{r^2 n \min_j p_j}{3}} = \delta$, then we have $r = \sqrt{\frac{3\ln\frac{2c}{\delta}}{n \min_j p_j}}$. Hence, the following holds with probability at least $1 - \delta$:

$$\sum_j |\frac{t_j}{n} - p_j| \leq \sqrt{\frac{3\ln\frac{2c}{\delta}}{n \min_j p_j}}. \tag{13}$$

Since $P\left(|\frac{t_j}{n} - p_j| \geq rp_j\right) \leq P\left(\cup_j \{|\frac{t_j}{n} - p_j| \geq rp_j\}\right)$, according to the proof process of inequality (13), we also have the following holds with probability at least $1 - \delta$:

$$|\frac{t_j}{n} - p_j| \leq p_j \sqrt{\frac{3\ln\frac{2c}{\delta}}{n \min_j p_j}}.$$

Solving the above inequality yields $t_j \geq np_j(1 - \sqrt{\frac{3\ln\frac{2c}{\delta}}{n \min_j p_j}})$. Similarly, we have $|\frac{u_j}{n} - (1 - p_j)| \leq (1 - p_j)\sqrt{\frac{3\ln\frac{2c}{\delta}}{n \min_j p_j}}$, and solving this inequality yields $u_j \geq n(1 - p_j)(1 - \sqrt{\frac{3\ln\frac{2c}{\delta}}{n \min_j p_j}})$.

Hence, we have the following holds with probability at least $1 - \delta$:

$$s_j = \min\{t_j, u_j\} \geq \min\{np_j(1 - \sqrt{\frac{3\ln\frac{2c}{\delta}}{n \min_j p_j}}), n(1 - p_j)(1 - \sqrt{\frac{3\ln\frac{2c}{\delta}}{n \min_j p_j}})\}. \tag{14}$$

In order to ensure that for every $j$-th label, disjoint positive and negative sample pairs can be constructed, we need to derive the lower bound for $\min_j\{s_j\}$ to obtain the number of disjoint positive and negative sample pairs that can be constructed for every $j$-th label. Since

$$s_j \geq \min_j\{s_j\} = \min_j\{\min\{t_j, u_j\}\}$$

$$\geq \min\{\min_j\{np_j(1 - \sqrt{\frac{3\ln\frac{2c}{\delta}}{n \min_j p_j}})\}, \min_j\{n(1 - p_j)(1 - \sqrt{\frac{3\ln\frac{2c}{\delta}}{n \min_j p_j}})\}\}$$

$$= \min\{n \min_j p_j(1 - \sqrt{\frac{3\ln\frac{2c}{\delta}}{n \min_j p_j}}), n \min_j(1 - p_j)(1 - \sqrt{\frac{3\ln\frac{2c}{\delta}}{n \min_j p_j}})\}$$

$$\geq \min\{\frac{1}{2}n \min_j p_j, \frac{1}{2}n \min_j(1 - p_j)\} \quad \text{(Assume that } n \geq \frac{12\ln\frac{2c}{\delta}}{\min_j p_j})$$

$$= \frac{1}{2}n \min\{\min_j p_j, \min_j(1 - p_j)\}$$

$$:= \frac{1}{2}s_0 \quad \text{(Define } s_0 = n \min\{\min_j p_j, \min_j(1 - p_j)\}),$$

then we have $s_j \geq \frac{1}{2}s_0$ with probability at least $1 - \delta$.

According to inequality (12) and Union bound Inequality, we have the following holds with probability at least $1 - \delta$:

$$\sum_j p_j \left| R(\boldsymbol{f}^{\boldsymbol{z}}|j) - \widehat{R}_D(\boldsymbol{f}^{\boldsymbol{z}}|j) \right| \leq \sum_j p_j \left( 2\Re_{s_j}(\mathcal{L}) + 8M\sqrt{\frac{2\ln\frac{c}{\delta}}{s_j}} \right) \tag{15}$$

Combining inequalities (8), (15), (13) and Union bound Inequality, we have the following holds with probability at least $1 - \delta$:

$$R(\boldsymbol{f}^{\boldsymbol{z}}) - \widehat{R}_D(\boldsymbol{f}^{\boldsymbol{z}})$$
$$\leq \sum_j p_j \left| R(\boldsymbol{f}^{\boldsymbol{z}}|j) - \widehat{R}_D(\boldsymbol{f}^{\boldsymbol{z}}|j) \right| + \sum_j |p_j - \frac{t_j}{n}|M$$
$$\leq \sum_j p_j \left( 2\Re_{s_j}(\mathcal{L}) + 8M\sqrt{\frac{2\ln\frac{2c}{\delta}}{s_j}} \right) + M\sqrt{\frac{3\ln\frac{4c}{\delta}}{n\min_j p_j}}$$
$$\leq 2\sum_j p_j \Re_{s_j}(\mathcal{L}) + 8M\sum_j p_j\sqrt{\frac{2\ln\frac{2c}{\delta}}{s_j}} + M\sqrt{\frac{3\ln\frac{4c}{\delta}}{n\min_j p_j}} \tag{16}$$

Next, we transform $\Re_{s_j}(\mathcal{L})$ in inequality (16) into $\hat{\Re}_{D_j}(\mathcal{L})$, according to McDiarmid's inequality, it is easy to obtain that the following holds with probability at least $1 - \delta$:

$$\Re_{s_j}(\mathcal{L}) \leq \hat{\Re}_{D_j}(\mathcal{L}) + M\sqrt{\frac{\ln\frac{1}{\delta}}{2s_j}}.$$

With Union bound Inequality, we have the following holds with probability at least $1 - \delta$:

$$\sum_j p_j \Re_{s_j}(\mathcal{L})$$
$$\leq \sum_j p_j \left( \hat{\Re}_{D_j}(\mathcal{L}) + M\sqrt{\frac{\ln\frac{c}{\delta}}{2s_j}} \right)$$
$$\leq \sum_j p_j \hat{\Re}_{D_j}(\mathcal{L}) + M\sum_j p_j\sqrt{\frac{\ln\frac{c}{\delta}}{2s_j}}. \tag{17}$$

Combining inequalities (16), (17), and Union bound Inequality, we have the following holds with probability at least $1 - \delta$:

$$R(\boldsymbol{f}^{\boldsymbol{z}}) - \widehat{R}_D(\boldsymbol{f}^{\boldsymbol{z}})$$
$$\leq 2\sum_j p_j \hat{\Re}_{D_j}(\mathcal{L}) + 2M\sum_j p_j\sqrt{\frac{\ln\frac{2c}{\delta}}{2s_j}} + 8M\sum_j p_j\sqrt{\frac{2\ln\frac{4c}{\delta}}{s_j}} + M\sqrt{\frac{3\ln\frac{8c}{\delta}}{n\min_j p_j}}$$
$$\leq 2\sqrt{2}\sum_j p_j \hat{\Re}_{D_0}(\mathcal{L}) + 2M\sum_j p_j\sqrt{\frac{\ln\frac{2c}{\delta}}{s_0}} + 8M\sum_j p_j\sqrt{\frac{4\ln\frac{4c}{\delta}}{s_0}} + M\sqrt{\frac{3\ln\frac{8c}{\delta}}{s_0}} \quad (\text{Use } s_j \geq \frac{1}{2}s_0 \text{ and } n\min_j p_j \geq s_0)$$
$$\leq 2\sqrt{2}\sum_j p_j \hat{\Re}_{D_0}(\mathcal{L}) + 22M\sum_j p_j\sqrt{\frac{\ln\frac{8c}{\delta}}{s_0}}, \tag{18}$$

where $\hat{\Re}_{D_0}(\mathcal{L}) = \mathbb{E}_{\boldsymbol{\epsilon}}\left[ \sup_{\boldsymbol{f}^{\boldsymbol{z}} \in \mathcal{F}} \frac{1}{s_0} \sum_{i=1}^{s_0} \epsilon_i \ell_R(\boldsymbol{f}^{\boldsymbol{z}}(\boldsymbol{x}_i, \boldsymbol{x}'_i)) \right] = \mathbb{E}_{\boldsymbol{\epsilon}}\left[ \sup_{\boldsymbol{f}^{\boldsymbol{z}} \in \mathcal{F}} \frac{1}{s_0} \sum_{i=1}^{s_0} \frac{1}{c} \sum_{j=1}^{c} \epsilon_i \ell_j \left( f_j^{\boldsymbol{z}}(\boldsymbol{x}_i) - f_j^{\boldsymbol{z}}(\boldsymbol{x}'_i) \right) \right].$

Then, according to Lemma 4.4 and Proposition 4.3, we have

$$\hat{\mathfrak{R}}_{D_0}(\mathcal{L}) \le \frac{12M}{\sqrt{s_0}} + 96\mu\sqrt{c}\widetilde{\mathfrak{R}}_{s_0 c}(\mathcal{P}(\mathcal{F})) \times (1 + \log_2(4es_0^2 c^2 \mu^2) \cdot \ln\frac{M\sqrt{s_0}}{\mu E}).$$

We then upper bound the worst-case Rademacher complexity $\widetilde{\mathfrak{R}}_{s_0 c}(\mathcal{P}(\mathcal{F}))$ as the following:

$$\widetilde{\mathfrak{R}}_{s_0 c}(\mathcal{P}(\mathcal{F}))$$

$$= \sup_{[c] \times D_0 \in [c] \times \mathcal{X}^{s_0}} \hat{\mathfrak{R}}_{[c] \times D_0}(\mathcal{P}(\mathcal{F}))$$

$$= \sup_{[c] \times D_0 \in [c] \times \mathcal{X}^{s_0}} \mathbb{E}_{\boldsymbol{\epsilon}} \left[ \sup_{p_j(\boldsymbol{f^z}(\boldsymbol{x}_i)) \in \mathcal{P}(\mathcal{F})} \frac{1}{s_0 c} \sum_{i=1}^{s_0} \sum_{j=1}^{c} \epsilon_{ij} p_j(\boldsymbol{f^z}(\boldsymbol{x}_i, \boldsymbol{x}_i')) \right]$$

$$= \sup_{[c] \times D_0 \in [c] \times \mathcal{X}^{s_0}} \mathbb{E}_{\boldsymbol{\epsilon}} \left[ \sup_{f_j^{\boldsymbol{z}} \in \mathcal{F}_j} \frac{1}{s_0 c} \sum_{i=1}^{s_0} \sum_{j=1}^{c} \epsilon_{ij} \left( f_j^{\boldsymbol{z}}(\boldsymbol{x}_i) - f_j^{\boldsymbol{z}}(\boldsymbol{x}_i') \right) \right]$$

$$= \sup_{\|\zeta_j(\phi_j(\boldsymbol{x}_i, \psi_j(\boldsymbol{z})))\|_2 \le A : i \in [s_0], j \in [c]} \frac{1}{s_0 c} \mathbb{E}_{\boldsymbol{\epsilon}} \left[ \sup_{\|\boldsymbol{w}_j\|_2 \le \Lambda} \sum_{i=1}^{s_0} \sum_{j=1}^{c} \epsilon_{ij} \langle \boldsymbol{w}_j, \zeta_j(\phi_j(\boldsymbol{x}_i, \psi_j(\boldsymbol{z}))) - \zeta_j(\phi_j(\boldsymbol{x}_i', \psi_j(\boldsymbol{z}))) \rangle \right]$$

$$= \sup_{\|\zeta_j(\phi_j(\boldsymbol{x}_i, \psi_j(\boldsymbol{z})))\|_2 \le A : i \in [s_0], j \in [c]} \frac{\Lambda}{s_0 c} \mathbb{E}_{\boldsymbol{\epsilon}} \| \sum_{i=1}^{s_0} \sum_{j=1}^{c} \epsilon_{ij} \left( \zeta_j(\phi_j(\boldsymbol{x}_i, \psi_j(\boldsymbol{z}))) - \zeta_j(\phi_j(\boldsymbol{x}_i', \psi_j(\boldsymbol{z}))) \right) \|$$

$$\le \sup_{\|\zeta_j(\phi_j(\boldsymbol{x}_i, \psi_j(\boldsymbol{z})))\|_2 \le A : i \in [s_0], j \in [c]} \frac{2\Lambda}{s_0 c} \mathbb{E}_{\boldsymbol{\epsilon}} \| \sum_{i=1}^{s_0} \sum_{j=1}^{c} \epsilon_{ij} \zeta_j(\phi_j(\boldsymbol{x}_i, \psi_j(\boldsymbol{z}))) \|$$

$$\le \sup_{\|\zeta_j(\phi_j(\boldsymbol{x}_i, \psi_j(\boldsymbol{z})))\|_2 \le A : i \in [s_0], j \in [c]} \frac{2\Lambda}{s_0 c} \left[ \mathbb{E}_{\boldsymbol{\epsilon}} \| \sum_{i=1}^{s_0} \sum_{j=1}^{c} \epsilon_{ij} \zeta_j(\phi_j(\boldsymbol{x}_i, \psi_j(\boldsymbol{z}))) \|^2 \right]^{\frac{1}{2}} \quad \text{(Use Jensen's Inequality)}$$

$$\le \sup_{\|\zeta_j(\phi_j(\boldsymbol{x}_i, \psi_j(\boldsymbol{z})))\|_2 \le A : i \in [s_0], j \in [c]} \frac{2\Lambda}{s_0 c} \left[ \sum_{i=1}^{s_0} \sum_{j=1}^{c} \|\zeta_j(\phi_j(\boldsymbol{x}_i, \psi_j(\boldsymbol{z})))\|^2 \right]^{\frac{1}{2}} \le \frac{2\Lambda A}{\sqrt{s_0 c}}. \quad \text{(Use Lemma A.3)} \tag{19}$$

Then, we have

$$\hat{\mathfrak{R}}_{D_0}(\mathcal{L}) \le \frac{12M}{\sqrt{s_0}} + \frac{192\mu\Lambda A \times (1 + \log_2(4es_0^2 c^2 \mu^2) \cdot \ln\frac{M\sqrt{s_0}}{\mu E})}{\sqrt{s_0}}.$$

Combining with (18), then

$$R(\boldsymbol{f^z}) \le \widehat{R}_D(\boldsymbol{f^z}) + \frac{24\sqrt{2}Mq}{\sqrt{s_0}} + \frac{384\sqrt{2}q\mu\Lambda A \times (1 + \log_2(4es_0^2 c^2 \mu^2) \cdot \ln\frac{M\sqrt{s_0}}{\mu E})}{\sqrt{s_0}} + 22Mq\sqrt{\frac{\ln\frac{8c}{\delta}}{s_0}},$$

where $q = \sum_j p_j$.

## B. General Bounds for Typical Controllable Learning Methods

### B.1. Proof of Theorem 5.1

**Proof Sketch**: First, according to the definition of $\ell_2$ norm, we have that the upper bound of $\|\phi_j(\boldsymbol{x}_i, \psi_j(\boldsymbol{z}))\|$ is $R + C$ for any $i \in [n]$, $j \in [c]$. Then, using Jensen's Inequality and Khintchine-Kahane inequality, the desired bound can be derived.

First, according to the definitions in Subsection 5.1 and $\ell_2$ norm, we have

$$\|\phi_j(\boldsymbol{x}, \psi_j(\boldsymbol{z}))\| := \|\text{Concate}(\boldsymbol{x}, \psi_\text{j}(\boldsymbol{z}))\| := \|\text{Concate}(\boldsymbol{x}, \boldsymbol{a}^\text{j})\|$$

$$=\|[x_1, \ldots, x_d, a_1^j, \ldots, a_{d'}^j]^\top\|$$

$$=\sqrt{\sum_{i=1}^{d} |x_i|^2 + \sum_{i'=1}^{d'} |a_{i'}^j|^2}$$

$$\leq\sqrt{\sum_{i=1}^{d} |x_i|^2} + \sqrt{\sum_{i'=1}^{d'} |a_{i'}^j|^2}$$

$$=\|\boldsymbol{x}\| + \|\boldsymbol{a}^j\|$$

$$=\|\boldsymbol{x}\| + \|\psi_j(\boldsymbol{z})\|$$

Since $\|\boldsymbol{x}_i\| \leq R, \|\psi_j(\boldsymbol{z})\| \leq C$ for any $i \in [n], j \in [c]$, we have that for any $i \in [n], j \in [c]$:

$$\|\phi_j(\boldsymbol{x}_i, \psi_j(\boldsymbol{z}))\| \leq R + C. \tag{20}$$

According to the proof of Theorem 4.7 (i.e., the first five equations in (7)), we have

$$\widetilde{\Re}_{nc}(\mathcal{P}(\mathcal{F})) = \sup_{\|\zeta_j(\phi_j(\boldsymbol{x}_i, \psi_j(\boldsymbol{z})))\|_2 \leq A : i \in [n], j \in [c]} \frac{\Lambda}{nc} \mathbb{E}_{\boldsymbol{\epsilon}} \| \sum_{i=1}^{n} \sum_{j=1}^{c} \epsilon_{ij} \zeta_j(\phi_j(\boldsymbol{x}_i, \psi_j(\boldsymbol{z}))) \| \tag{21}$$

Then, we upper bound the worst-case Rademacher complexity $\widetilde{\Re}_{nc}(\mathcal{P}(\mathcal{F}))$ as the following:

$$\widetilde{\Re}_{nc}(\mathcal{P}(\mathcal{F}))$$

$$= \sup_{\|\zeta_j(\phi_j(\boldsymbol{x}_i, \psi_j(\boldsymbol{z})))\|_2 \leq A : i \in [n], j \in [c]} \frac{\Lambda}{nc} \mathbb{E}_{\boldsymbol{\epsilon}} \| \sum_{i=1}^{n} \sum_{j=1}^{c} \epsilon_{ij} \zeta_j(\phi_j(\boldsymbol{x}_i, \psi_j(\boldsymbol{z}))) \|$$

$$\leq \sup_{\|\phi_j(\boldsymbol{x}_i, \psi_j(\boldsymbol{z}))\|_2 \leq B : i \in [n], j \in [c]} \frac{2\Lambda\rho}{nc} \mathbb{E}_{\boldsymbol{\epsilon}} \| \sum_{i=1}^{n} \sum_{j=1}^{c} \epsilon_{ij} \phi_j(\boldsymbol{x}_i, \psi_j(\boldsymbol{z})) \| \quad \text{(Use } \zeta_j \text{ is } \rho\text{-Lipschitz)}$$

$$\leq \sup_{\|\phi_j(\boldsymbol{x}_i, \psi_j(\boldsymbol{z}))\|_2 \leq B : i \in [n], j \in [c]} \frac{2\rho\Lambda}{nc} \left[ \mathbb{E}_{\boldsymbol{\epsilon}} \| \sum_{i=1}^{n} \sum_{j=1}^{c} \epsilon_{ij} \phi_j(\boldsymbol{x}_i, \psi_j(\boldsymbol{z})) \|^2 \right]^{\frac{1}{2}} \quad \text{(Use Jensen's Inequality)}$$

$$\leq \sup_{\|\phi_j(\boldsymbol{x}_i, \psi_j(\boldsymbol{z}))\|_2 \leq B : i \in [n], j \in [c]} \frac{2\rho\Lambda}{nc} \left[ \sum_{i=1}^{n} \sum_{j=1}^{c} \|\phi_j(\boldsymbol{x}_i, \psi_j(\boldsymbol{z}))\|^2 \right]^{\frac{1}{2}} \quad \text{(Use Lemma A.3)}$$

$$\leq \frac{2\rho\Lambda(R+C)}{\sqrt{nc}}. \quad \text{(Use inequality (20))} \tag{22}$$

## B.2. Proof of Theorem 5.2

**Proof Sketch**: Since hypernetwork-based controllable learning method is two-stage, the hypernetwork is used to generate a vector of weights for each task target in the first stage, and then these generated parameter vectors are used in the second stage learning. Therefore, the corresponding whole function class is actually denoted as $\mathcal{H} = \mathcal{F} + \mathcal{L}_{\mu_\text{ctrl}} \circ \mathcal{G}^{\otimes c}$. Since the generated parameter vectors in the first stage are actually used as fixed parameters rather than inputs in the second stage, in order to fully consider the capacity of the hypernetwork corresponding to the first stage, it is necessary and appropriate to define the whole function class as the sum of the function classes $\mathcal{F} + \mathcal{L}_{\mu_\text{ctrl}} \circ \mathcal{G}^{\otimes c}$ corresponding to the models of these two stages. Then, the upper bound of the worst-case Rademacher complexity of the whole function class is transformed into upper bounding the worst-case Rademacher complexity $\widetilde{\Re}_{nc}(\mathcal{P}(\mathcal{F}))$ and $\widetilde{\Re}_{nc}(\mathcal{L}_{\mu_\text{ctrl}} \circ \mathcal{P}(\mathcal{G}^{\otimes c}))$ respectively. Finally, using Jensen's Inequality and Khintchine-Kahane inequality, the desired upper bounds can be derived respectively.

Since hypernetwork-based controllable learning method is two-stage, the hypernetwork is used to generate a vector of weights for each task target in the first stage, and then the generated parameter vectors are used in the second stage learning. Therefore, the corresponding whole function class is actually denoted as $\mathcal{H} = \mathcal{F} + \mathcal{L}_{\text{ctrl}} \circ \mathcal{G}^{\otimes c}$. Then, we have

$$
\begin{aligned}
&\widetilde{\Re}_{nc}(\mathcal{P}(\mathcal{H})) \\
=&\widetilde{\Re}_{nc}(\mathcal{P}(\mathcal{F} + \mathcal{L}_{\text{ctrl}} \circ \mathcal{G}^{\otimes c})) \\
=&\widetilde{\Re}_{nc}(\mathcal{P}(\mathcal{F}) + \mathcal{P}(\mathcal{L}_{\text{ctrl}} \circ \mathcal{G}^{\otimes c})) \\
=&\widetilde{\Re}_{nc}(\mathcal{P}(\mathcal{F}) + \mathcal{L}_{\text{ctrl}} \circ \mathcal{P}(\mathcal{G}^{\otimes c})) \\
\leq&\widetilde{\Re}_{nc}(\mathcal{P}(\mathcal{F})) + \widetilde{\Re}_{nc}(\mathcal{L}_{\text{ctrl}} \circ \mathcal{P}(\mathcal{G}^{\otimes c})). \quad \text{(Use Theorem 12 in (Bartlett \& Mendelson, 2002))}
\end{aligned}
$$

Hence, the upper bound of the worst-case Rademacher complexity for hypernetwork-based controllable learning methods is transformed into upper bounding the worst-case Rademacher complexity $\widetilde{\Re}_{nc}(\mathcal{P}(\mathcal{F}))$ and $\widetilde{\Re}_{nc}(\mathcal{L}_{\text{ctrl}} \circ \mathcal{P}(\mathcal{G}^{\otimes c}))$ respectively.

We first upper bound the worst-case Rademacher complexity $\widetilde{\Re}_{nc}(\mathcal{P}(\mathcal{F}))$ as the following:

$$
\begin{aligned}
\widetilde{\Re}_{nc}(\mathcal{P}(\mathcal{F})) =& \sup_{\|\zeta_j(\phi_j(\boldsymbol{x}_i;\boldsymbol{w}_{\psi_j}))\|_2 \leq A : i \in [n], j \in [c]} \frac{\Lambda}{nc} \mathbb{E}_{\boldsymbol{\epsilon}} \| \sum_{i=1}^{n} \sum_{j=1}^{c} \epsilon_{ij} \zeta_j(\phi_j(\boldsymbol{x}_i;\boldsymbol{w}_{\psi_j})) \| \quad \text{(Use inequality (21))} \\
\leq& \sup_{\|\phi_j(\boldsymbol{x}_i;\boldsymbol{w}_{\psi_j})\|_2 \leq B : i \in [n], j \in [c]} \frac{2\Lambda\rho}{nc} \mathbb{E}_{\boldsymbol{\epsilon}} \| \sum_{i=1}^{n} \sum_{j=1}^{c} \epsilon_{ij} \phi_j(\boldsymbol{x}_i;\boldsymbol{w}_{\psi_j}) \| \quad \text{(Use $\zeta_j$ is $\rho$-Lipschitz)} \\
\leq& \sup_{\|\phi_j(\boldsymbol{x}_i;\boldsymbol{w}_{\psi_j})\|_2 \leq B : i \in [n], j \in [c]} \frac{2\rho\Lambda}{nc} \left[ \mathbb{E}_{\boldsymbol{\epsilon}} \| \sum_{i=1}^{n} \sum_{j=1}^{c} \epsilon_{ij} \phi_j(\boldsymbol{x}_i;\boldsymbol{w}_{\psi_j}) \|^2 \right]^{\frac{1}{2}} \quad \text{(Use Jensen's Inequality)} \\
\leq& \sup_{\|\phi_j(\boldsymbol{x}_i;\boldsymbol{w}_{\psi_j})\|_2 \leq B : i \in [n], j \in [c]} \frac{2\rho\Lambda}{nc} \left[ \sum_{i=1}^{n} \sum_{j=1}^{c} \| \phi_j(\boldsymbol{x}_i;\boldsymbol{w}_{\psi_j}) \|^2 \right]^{\frac{1}{2}} \leq \frac{2\rho\Lambda B}{\sqrt{nc}}. \quad \text{(Use Lemma A.3)} \quad (23)
\end{aligned}
$$

We then upper bound the worst-case Rademacher complexity $\widetilde{\Re}_{nc}(\mathcal{L}_{\text{ctrl}} \circ \mathcal{P}(\mathcal{G}^{\otimes c}))$ as the following:

$$
\begin{aligned}
&\widetilde{\Re}_{nc}(\mathcal{L}_{\text{ctrl}} \circ \mathcal{P}(\mathcal{G}^{\otimes c})) \\
=& \sup_{[c] \times Z \in [c] \times \mathcal{Z}^n} \mathbb{E}_{\boldsymbol{\epsilon}} \left[ \sup_{\ell_{\text{ctrl}}(\psi_j(\boldsymbol{z})) \in \mathcal{L}_{\text{ctrl}} \circ \mathcal{G}_j} \frac{1}{nc} \sum_{i=1}^{n} \sum_{j=1}^{c} \epsilon_{ij} \ell_{\text{ctrl}}(\psi_j(\boldsymbol{z}_i)) \right] \\
\leq& \sup_{\psi_j(\boldsymbol{z}_i) \in \mathcal{G}_j} 2\mu_{\text{ctrl}} \mathbb{E}_{\boldsymbol{\epsilon}} \left[ \frac{1}{nc} \sum_{i=1}^{n} \sum_{j=1}^{c} \epsilon_{ij} \psi_j(\boldsymbol{z}_i) \right] \quad \text{(Use $\ell_{\text{ctrl}}$ is $\mu_{\text{ctrl}}$-Lipschitz)} \\
\leq& \sup_{|\psi_j(\boldsymbol{z}_i)| \leq C : i \in [n], j \in [c]} \frac{2\mu_{\text{ctrl}}}{nc} \left[ \sum_{i=1}^{n} \sum_{j=1}^{c} \psi_j(\boldsymbol{z}_i)^2 \right]^{\frac{1}{2}} \quad \text{(Use Lemma A.3)} \\
\leq& \frac{2\mu_{\text{ctrl}} C}{\sqrt{nc}}.
\end{aligned}
$$

Hence, we can derive the upper bound of the worst-case Rademacher complexity for hypernetwork-based controllable learning methods:

$$
\widetilde{\Re}_{nc}(\mathcal{P}(\mathcal{H})) \leq \frac{2\rho\Lambda B}{\sqrt{nc}} + \frac{2\mu_{\text{ctrl}} C}{\sqrt{nc}}.
$$

## B.3. Proof of Theorem 5.4

We upper bound the Rademacher complexity $\hat{\mathfrak{R}}_D(\mathcal{F})$ of 5 layer FNN as the following:

$$\hat{\mathfrak{R}}_D(\mathcal{F}) = \mathbb{E}_{\boldsymbol{\epsilon}} \left[ \sup_{f \in \mathcal{F}} \frac{1}{n} \sum_{i=1}^{n} \epsilon_i f\left(\boldsymbol{x}_i\right) \right]$$

$$= \mathbb{E}_{\boldsymbol{\epsilon}} \left[ \sup_{\|\boldsymbol{w}\| \leq \Lambda, \sigma} \frac{1}{n} \sum_{i=1}^{n} \epsilon_i \boldsymbol{w}^\top \sigma(W_5 \sigma(W_4 \cdots \sigma(W_1 \boldsymbol{x}_i))) \right]$$

$$\leq \Lambda \mathbb{E}_{\boldsymbol{\epsilon}} \sup_{\sigma} \frac{1}{n} \| \sum_{i=1}^{n} \epsilon_i \sigma(W_5 \sigma(W_4 \cdots \sigma(W_1 \boldsymbol{x}_i))) \|$$

$$\leq 2 \Lambda \mathbb{E}_{\boldsymbol{\epsilon}} \sup_{\|W_5\| \leq B_F, \sigma} \frac{1}{n} \| \sum_{i=1}^{n} \epsilon_i W_5 \sigma(W_4 \cdots \sigma(W_1 \boldsymbol{x}_i)) \| \quad \text{(ReLU activation is 1-Lipschitz)}$$

$$\leq 2 \Lambda B_F \mathbb{E}_{\boldsymbol{\epsilon}} \sup_{\sigma} \frac{1}{n} \| \sum_{i=1}^{n} \epsilon_i \sigma(W_4 \cdots \sigma(W_1 \boldsymbol{x}_i)) \|$$

$$\leq 2^2 \Lambda B_F \mathbb{E}_{\boldsymbol{\epsilon}} \sup_{\|W_4\| \leq B_F, \sigma} \frac{1}{n} \| \sum_{i=1}^{n} \epsilon_i W_4 \sigma(W_3 \cdots \sigma(W_1 \boldsymbol{x}_i)) \|$$

$$\leq 2^2 \Lambda B_F^2 \mathbb{E}_{\boldsymbol{\epsilon}} \sup_{\sigma} \frac{1}{n} \| \sum_{i=1}^{n} \epsilon_i \sigma(W_3 \sigma(W_2 \sigma(W_1 \boldsymbol{x}_i))) \|$$

$$\leq 2^3 \Lambda B_F^2 \mathbb{E}_{\boldsymbol{\epsilon}} \sup_{\|W_3\| \leq B_F, \sigma} \frac{1}{n} \| \sum_{i=1}^{n} \epsilon_i W_3 \sigma(W_2 \sigma(W_1 \boldsymbol{x}_i)) \|$$

$$\leq 2^3 \Lambda B_F^3 \mathbb{E}_{\boldsymbol{\epsilon}} \sup_{\sigma} \frac{1}{n} \| \sum_{i=1}^{n} \epsilon_i \sigma(W_2 \sigma(W_1 \boldsymbol{x}_i)) \|$$

$$\leq 2^4 \Lambda B_F^3 \mathbb{E}_{\boldsymbol{\epsilon}} \sup_{\|W_2\| \leq B_F, \sigma} \frac{1}{n} \| \sum_{i=1}^{n} \epsilon_i W_2 \sigma(W_1 \boldsymbol{x}_i) \|$$

$$\leq 2^4 \Lambda B_F^4 \mathbb{E}_{\boldsymbol{\epsilon}} \sup_{\sigma} \frac{1}{n} \| \sum_{i=1}^{n} \epsilon_i \sigma(W_1 \boldsymbol{x}_i) \|$$

$$\leq 2^5 \Lambda B_F^4 \mathbb{E}_{\boldsymbol{\epsilon}} \sup_{\|W_1\| \leq B_F} \frac{1}{n} \| \sum_{i=1}^{n} \epsilon_i W_1 \boldsymbol{x}_i \|$$

$$\leq 2^5 \Lambda B_F^5 \mathbb{E}_{\boldsymbol{\epsilon}} \frac{1}{n} \| \sum_{i=1}^{n} \epsilon_i \boldsymbol{x}_i \|$$

$$\leq \frac{2^5 \Lambda B_F^5}{n} \left[ \mathbb{E}_{\boldsymbol{\epsilon}} \| \sum_{i=1}^{n} \epsilon_i \boldsymbol{x}_i \|^2 \right]^{\frac{1}{2}} \quad \text{(Use Jensen's Inequality)}$$

$$\leq \frac{2^5 \Lambda B_F^5}{n} \left[ \sum_{i=1}^{n} \| \boldsymbol{x}_i \|^2 \right]^{\frac{1}{2}} \quad \text{(Use Lemma A.3)}$$

$$\leq \frac{2^5 \Lambda B_F^5 R}{\sqrt{n}}.$$

## B.4. Proof of Theorem 5.6

We first introduce the following lemma:

**Lemma B.1** (Corollary A.7 in (Edelman et al., 2022)). *For vectors* $\theta_1, \theta_2 \in \mathbb{R}^p$,

$$\|\text{softmax}\left(\theta_1\right) - \text{softmax}\left(\theta_2\right)\|_1 \leq 2 \|\theta_1 - \theta_2\|_\infty.$$

Then, we upper bound the Rademacher complexity $\hat{\Re}_D(\mathcal{F})$ of the scalar one layer Transformer as the following:

$$\hat{\Re}_D(\mathcal{F}) = \mathbb{E}_{\boldsymbol{\epsilon}}\left[\sup_{\boldsymbol{w}^\top \boldsymbol{y}_{[\text{CLS}]} \in \mathcal{F}} \frac{1}{n}\sum_{i=1}^n \epsilon_i \boldsymbol{w}^\top W_C^\top \sigma\left(W_V^\top X_i^\top \text{softmax}\left(X_i W_{QK}^\top \boldsymbol{x}_{[\text{CLS}]}\right)\right)\right]$$

$$=\mathbb{E}_{\boldsymbol{\epsilon}}\left[\sup_{\|\boldsymbol{w}\|\leq B_{\boldsymbol{w}},\sigma} \frac{1}{n}\sum_{i=1}^n \epsilon_i \langle \boldsymbol{w}, W_C^\top \sigma\left(W_V^\top X_i^\top \text{softmax}\left(X_i W_{QK}^\top \boldsymbol{x}_{[\text{CLS}]}\right)\right)\rangle\right]$$

$$\leq B_{\boldsymbol{w}}\mathbb{E}_{\boldsymbol{\epsilon}}\sup_{\|W_C\|\leq B_{W_C},\sigma} \frac{1}{n}\|\sum_{i=1}^n \epsilon_i W_C^\top \sigma\left(W_V^\top X_i^\top \text{softmax}\left(X_i W_{QK}^\top \boldsymbol{x}_{[\text{CLS}]}\right)\right)\|$$

$$\leq \frac{B_{\boldsymbol{w}}B_{W_C}}{n}\mathbb{E}_{\boldsymbol{\epsilon}}\sup_\sigma \|\sum_{i=1}^n \epsilon_i \sigma\left(W_V^\top X_i^\top \text{softmax}\left(X_i W_{QK}^\top \boldsymbol{x}_{[\text{CLS}]}\right)\right)\|$$

$$\leq \frac{2B_{\boldsymbol{w}}B_{W_C}L_\sigma}{n}\mathbb{E}_{\boldsymbol{\epsilon}}\sup_{W_V,W_{QK}} \|\sum_{i=1}^n \epsilon_i W_V^\top X_i^\top \text{softmax}\left(X_i W_{QK}^\top \boldsymbol{x}_{[\text{CLS}]}\right)\|$$

$$\leq \frac{2B_{\boldsymbol{w}}B_{W_C}B_{W_V}L_\sigma}{n}\mathbb{E}_{\boldsymbol{\epsilon}}\sup_{W_{QK}} \|\sum_{i=1}^n \epsilon_i X_i^\top \text{softmax}\left(X_i W_{QK}^\top \boldsymbol{x}_{[\text{CLS}]}\right)\|$$

$$\leq \frac{2B_{\boldsymbol{w}}B_{W_C}B_{W_V}L_\sigma}{n}\mathbb{E}_{\boldsymbol{\epsilon}}\sup_{i,W_{QK}} \|\sum_{i=1}^n \epsilon_i X_i^\top \|_{2,\infty}\|\text{softmax}\left(X_i W_{QK}^\top \boldsymbol{x}_{[\text{CLS}]}\right)\|_1 \quad (\text{Use } \|P\boldsymbol{x}\|\leq\|P\|_{2,\infty}\|\boldsymbol{x}\|_1)$$

$$\leq \frac{2B_{\boldsymbol{w}}B_{W_C}B_{W_V}L_\sigma}{n}\sup_{i,W_{QK}} \|\text{softmax}\left(X_i W_{QK}^\top \boldsymbol{x}_{[\text{CLS}]}\right)\|_1 \mathbb{E}_{\boldsymbol{\epsilon}}\|\sum_{i=1}^n \epsilon_i X_i^\top \|_{2,\infty}$$

$$\leq \frac{4B_{\boldsymbol{w}}B_{W_C}B_{W_V}L_\sigma}{n}\sup_{i,W_{QK}} \|X_i W_{QK}^\top \boldsymbol{x}_{[\text{CLS}]}\|_\infty \mathbb{E}_{\boldsymbol{\epsilon}}\|\sum_{i=1}^n \epsilon_i X_i^\top \|_{2,\infty} \quad (\text{Use Lemma B.1})$$

$$= \frac{4B_{\boldsymbol{w}}B_{W_C}B_{W_V}L_\sigma}{n}\sup_{i,W_{QK}}\max_t \|\boldsymbol{x}_t^{i\top} W_{QK}^\top \boldsymbol{x}_{[\text{CLS}]}\|\mathbb{E}_{\boldsymbol{\epsilon}}\|\sum_{i=1}^n \epsilon_i X_i^\top \|_{2,\infty}$$

$$\leq \frac{4B_{\boldsymbol{w}}B_{W_C}B_{W_V}L_\sigma}{n}\sup_{i,W_{QK}}\max_t \|W_{QK}\boldsymbol{x}_t^i\|\|\boldsymbol{x}_{[\text{CLS}]}\|\mathbb{E}_{\boldsymbol{\epsilon}}\|\sum_{i=1}^n \epsilon_i X_i^\top \|_{2,\infty}$$

$$\leq \frac{4B_{\boldsymbol{w}}B_{W_C}B_{W_V}B_{W_{QK}}L_\sigma}{n}\max_{i,t}\|\boldsymbol{x}_t^i\|\|\boldsymbol{x}_{[\text{CLS}]}\|\mathbb{E}_{\boldsymbol{\epsilon}}\|\sum_{i=1}^n \epsilon_i X_i^\top \|_{2,\infty}$$

$$\leq \frac{4B_{\boldsymbol{w}}B_{W_C}B_{W_V}B_{W_{QK}}L_\sigma R^2}{n}\mathbb{E}_{\boldsymbol{\epsilon}}\|\sum_{i=1}^n \epsilon_i X_i^\top \|_{2,\infty}$$

$$= \frac{4B_{\boldsymbol{w}}B_{W_C}B_{W_V}B_{W_{QK}}L_\sigma R^2}{n}\max_t \mathbb{E}_{\boldsymbol{\epsilon}}\|\sum_{i=1}^n \epsilon_i \boldsymbol{x}_t^i\|$$

$$\leq \frac{4B_{\boldsymbol{w}}B_{W_C}B_{W_V}B_{W_{QK}}L_\sigma R^2}{n}\max_t \left[\mathbb{E}_{\boldsymbol{\epsilon}}\|\sum_{i=1}^n \epsilon_i \boldsymbol{x}_t^i\|^2\right]^{\frac{1}{2}} \quad (\text{Use Jensen's Inequality})$$

$$\leq \frac{4B_{\boldsymbol{w}}B_{W_C}B_{W_V}B_{W_{QK}}L_\sigma R^2}{n}\max_t \left[\sum_{i=1}^n \|\boldsymbol{x}_t^i\|^2\right]^{\frac{1}{2}} \quad (\text{Use Lemma A.3})$$

$$\leq \frac{4B_{\boldsymbol{w}}B_{W_C}B_{W_V}B_{W_{QK}}L_\sigma R^3}{\sqrt{n}}.$$

