# OpenReview forum: "Generalization Analysis for Controllable Learning"
_ICML.cc/2025/Conference — ICML 2025 poster_

### Official Review · Reviewer_9TGj · 2025-03-02

**Overall Recommendation:** 4

**Summary:**

The paper establishes a statistical generalization theory for controllable learning, which enables machine learning models to dynamically adapt to task requirements during testing. This adaptability is crucial in many real-world applications of decision systems, such as recommender systems and information retrieval, where models need to accommodate diverse user preferences and changing objectives. Despite its empirical success, the generalization performance of controllable learning methods has not been comprehensively analyzed.

To fill this gap, the authors propose a unified theoretical framework for controllable learning and introduce a vector-contraction inequality, which yields the tightest known generalization bounds. These bounds are largely independent of the number of task targets, except for logarithmic factors. The key mathematical technique involves the development of a novel vector-contraction inequality. The paper then derives generalization bounds for two common controllable learning methods: embedding-based methods, which adjust model inputs, and hypernetwork-based methods, which generate task-specific model parameters. Additionally, the authors present modular theoretical results for frequently used control and prediction functions, offering broad theoretical guarantees for controllable learning methods. The findings enhance the understanding of controllability in machine learning and provide new insights for further research.

**Claims And Evidence:**

I think the claims are supported by theoretical results and there is no concern in my opinion regarding the theoretical results in this paper.

**Essential References Not Discussed:**

This paper describes two major areas of empirical works in controllable machine learning: the embedding based method where the specific tasks are mapped into embeddings, and hypernetwork based method which generates new model parameters on given task requirements.

**Experimental Designs Or Analyses:**

N/A

**Methods And Evaluation Criteria:**

N/A. This work is entirely theoretical, which provides a new statistical/mathematical theory for the generalization of controllable machine learning.

**Other Comments Or Suggestions:**

I believe this paper makes a valid contribution to generalization theory. However, there is room for improvement.

I would suggest including a discussion on the major technical challenges in the theoretical analysis. This would clarify what has hindered previous works from establishing a generalization theory for controllable learning using conventional analytical tools. For instance, the discussion could briefly explain why the number of tasks poses difficulties for traditional analysis methods. While the discussion does not need to be long, it should explicitly highlight the obstacles that have made developing a generalization theory for controllable learning challenging and how this paper overcomes them. This would help reinforce the paper's contribution, which is particularly important for a work that claims to be the first theoretical study of this learning method.

**Other Strengths And Weaknesses:**

Strength:

**1** Clarity: This paper is clearly written, effectively conveying its main ideas. The structure is well-organized, with general theoretical results presented in Section 4 and their application to specific machine learning frameworks in Section 5. Overall, the paper is easy to follow.

**2** Originality: The paper claims to be the first theoretical work on the generalization of controllable learning.

Weakness:

**1** Technical challenge and significance: It is not immediately clear what major technical challenges were encountered in developing the theory in this work. While the paper presents the first generalization theory for controllable learning, this alone does not sufficiently demonstrate that it tackled and resolved a significant technical difficulty.

**Questions For Authors:**

Please see my suggestion regarding major technical novelty in the above session **Comments or Suggestions**.

**Relation To Broader Scientific Literature:**

This paper falls into the greater scientific area of statistical/mathematical theory for generalization of machine learning models and methods. To this end, this paper first develops a new mathematical theorem which is a vector contraction inequality for Rademacher complexity (i.e. Lemma 4.4). It appears to me that this contribution is of independent interest and might be applied to the theoretical analysis of controllable learning by other works. Then, based on this tool, a unified theory on controllable learning is derived in Section 4, which is another potentially extendable theoretical technique.

**Theoretical Claims:**

Yes. I read the proof of Lemma 4.4 (the contraction inequality) in Appendix A.3.2, which is claimed to be the main technical tool for theoretical analysis. The proof looks valid. I also look at the proof (sketch) for Theorem 4.5 and 4.7.

I did not check all the details of the proof of the other lemmas in the appendix though.

---

> ### Author Rebuttal · Authors · 2025-03-31
>
> Thank you for your constructive comments and active interest in helping us improve the quality of the paper.
>
> The following are our responses to the Questions:
>
> **1. Response to Suggestions.**
>
> The difficulty of theoretical analysis for controllable learning lies in two aspects.
>
> First, the development of controllable learning is driven by real-world applications, and the methods developed are very different. It is difficult to establish a unified theoretical framework to cover all these typical methods. The lack of a unified framework is the most intuitive and primary obstacle to using existing analytical tools to establish an effective theory bound for controllable learning. In addition, controllable learning needs to dynamically and adaptively respond to task requirements and may involve many completely different task targets, which increases the challenge of explicitly reflecting these factors in generalization analysis. In addition, an effective unified theoretical framework should also be extensible and provide interfaces that can cover a wider variety of methods that may be potentially developed in the future. The definition of the controllable learning function class and the modular theoretical results provided for commonly used control functions and prediction functions in controllable learning ensure the establishment of a unified theoretical framework.
>
> Second, about reducing the dependency of the bounds on the number of task targets $c$. The analysis of the number of task targets in the bounds can be traced back to a basic bound with a linear dependency on $c$, which comes from the following inequality:
>
> $$\mathbb{E}\left[\sup_{\boldsymbol{f} \in \mathcal{F}} \frac{1}{n} \sum_{i=1}^n \sum_{j=1}^c \epsilon_{ij} f_j\left(\boldsymbol{x}_{i}\right)\right]  $$
>
> $$\leq  c \max_j \mathbb{E}\left[\sup_{f_j} \frac{1}{n} \sum_{i=1}^n  \epsilon_{ij} f_j\left(\boldsymbol{x}_{i}\right) \right].$$
>
> The dependency of the bounds on $c$ can be improved to square-root or logarithmic by preserving the coupling among different components reflected by the constraint, i.e., $\\|\boldsymbol{w}\\| \leq \Lambda$. However, each task target in controllable learning can be completely different, hence the coupling relationship between the weights of the functions corresponding to each task target needs to be decoupled. Hence, we introduce the projection operator. We found that the square-root dependency of the bound on $c$ is inevitable for $\ell_2$ Lipschitz loss, which essentially comes from the $\sqrt{c}$ factor in the radius of the empirical $\ell_2$ cover of the projection function class, but the Lipschitz continuity of the loss function with respect to $\ell_\infty$ norm can eliminate it. Hence, the tight bounds with no dependency on $c$, up to logarithmic terms, can be derived.

---

> > ### Comment · Reviewer_9TGj · 2025-04-04
> >
> > I thank the authors for their rebuttal. The rebuttal addressed my concerns and I updated my rating accordingly.
> >
> > Best,
> > reviewer

---

> > > ### Author Response · Authors · 2025-04-07
> > >
> > > Dear Reviewer 9TGj,
> > >
> > > Thank you for your insightful feedback, for your constructive comments, and for updating your score.
> > >
> > > Best regards,
> > >
> > > Authors

---

### Official Review · Reviewer_Rc3E · 2025-03-07

**Overall Recommendation:** 4

**Summary:**

This paper investigates the theory bounds of controllable learning methods, highlighting the need for a deeper understanding of controllable learning methods from a generalization perspective. This paper first gives a formal definition of the general function classes of controllable learning, then develops a novel vector-contraction inequality and derives tight generalization bounds for the general function classes of controllable learning. In addition, this paper analyzes two typical controllable learning methods and derives corresponding generalization bounds for specific methods. These theoretical results reveal the impact of different manipulation functions and control functions on the generalization bounds.

## update after rebuttal

**Claims And Evidence:**

This paper investigates the generalization of controllable learning methods for the first time, establishes a theoretical framework for controllable learning, and derives a series of tight generalization bounds. These theoretical results provide strong evidence for the claim that different manipulation and control functions will affect the constants in the generalization bounds.

**Essential References Not Discussed:**

All the essential related works have been adequately discussed.

**Experimental Designs Or Analyses:**

Yes. The analyses in this paper are complete and all the analyses are valid.

**Methods And Evaluation Criteria:**

Yes. The proposed theoretical analysis method can produce tight theory bounds.

**Other Comments Or Suggestions:**

Typos: Line 421, right column, "explanation the models" --> "explanation for the models".

**Other Strengths And Weaknesses:**

Strengths:

1. This paper is the first work to theoretically analyze controllability. It not only provides a basic framework for the theoretical analysis of controllable learning methods, but also provides some general analysis methods and theoretical tools applicable to typical controllable learning methods, which is a key contribution to a deeper understanding of controllability in machine learning.

2. This paper introduces many valuable theoretical results and techniques, among which the proposed novel vector-contraction inequality may have potential applications in a wider range of problem settings. In addition, the capacity-based theoretical results on deep neural networks can also provide deeper insights into understanding the role of these models in controllable learning methods.

3. The theoretical results in this paper are all rigorously proved, and a lot of analysis and explanations make the theoretical results easy to understand. The assumptions corresponding to the theoretical results are reasonable and consistent with the practical situations, and the argumentation process is logically coherent.

Weaknesses:

1. I noticed that the theoretical results in Section 5 are specific to the weighted Hamming loss, but the multi-target bipartite ranking loss is also analyzed in Section 4. Can the theoretical results in Section 5 be directly applied to the multi-target bipartite ranking loss?

2. Although the theoretical results presented in this paper are important, there seems to be a lack of discussion on how to use these theoretical results to guide the design of controllable learning methods in real scenarios.

**Questions For Authors:**

1. Can these theoretical results provide effective guidance for model settings of different modules of controllable learning methods in practical situations?

2. The assumptions corresponding to the proposed vector-contraction inequality seem to be general. Can this lemma be applied to theoretical analysis of other problem settings? Or can it be applied to other loss functions?

**Relation To Broader Scientific Literature:**

Previous research on controllability mainly focused on methods in information retrieval and recommender systems, and the related research lacked theoretical analysis. This work is an important step towards theoretical understanding of the controllability of machine learning from a generalization perspective, and it also explains why controllable learning methods have good generalization properties.

**Theoretical Claims:**

Yes. I have reviewed the proofs and the theoretical claims are correct.

---

> ### Author Rebuttal · Authors · 2025-03-31
>
> Thank you for your constructive comments and active interest in helping us improve the quality of the paper.
>
> The following are our responses to the Questions:
>
> **1. Response to the Weakness 1.**
>
> The theoretical results in Section 5 can be used on the multi-target bipartite ranking loss. When the specific controllable learning method involves the multi-target bipartite ranking loss, we only need to consider replacing the corresponding constants in Theorem 4.7 (instead of Theorem 4.5) as in the proof process in Subsection 5.3.
>
> **2. Response to the Weakness 2 and Question 1.**
>
> Controllable learning often involves the selection of control functions and prediction functions corresponding to task targets. Although increasing the capacity of the model can enhance the representation ability of controllable learning, the increase in capacity is not blind. Our theoretical results show that for embedding-based controllable learning methods, the control function should choose to use a large-capacity model, and for hypernetwork-based controllable learning methods, the main model used for classification (rather than the hypernetwork) should choose to use a large-capacity model. These theoretical results are consistent with practical methods and can serve as a general guide for the structural design of controllable learning models.
>
> **3. Response to the Question 2.**
>
> The proposed vector-contraction inequality only assumes that the loss is Lipschitz continuous with respect to the $\ell_\infty$ norm. In addition, many loss functions in other problem settings also satisfy this assumption, such as multi-class classification [1], [2], multi-label learning [3], clustering [4], etc., which means that our theoretical results can also be extended when it comes to dealing with these problems, which also shows that the assumption on the loss function do not lose generality.
>
> [1] Maksim Lapin, Matthias Hein, Bernt Schiele. "Top-k Multiclass SVM", NIPS 2015.
>
> [2] Yunwen Lei, Ürün Dogan, Ding-Xuan Zhou, Marius Kloft. "Data-dependent generalization bounds for multi-class classification", IEEE TIT 2019.
>
> [3] Yi-Fan Zhang, Min-Ling Zhang. "Generalization Analysis for Multi-Label Learning", ICML 2024.
>
> [4] Shaojie Li, Yong Liu. "Sharper Generalization Bounds for Clustering", ICML 2021.

---

### Official Review · Reviewer_Gt3V · 2025-03-07

**Overall Recommendation:** 4

**Summary:**

This paper analyzes the generalization of controllable learning methods to understand controllability in trustworthy machine learning better. It establishes a unified and practical framework for the theoretical study of controllable learning methods and proposes a novel vector-contraction inequality that can derive tight generalization bounds for them. In addition, it derives generalization bounds for two typical controllable learning methods: embedding-based and hypernetwork methods. These results reveal the impact of different manipulation methods based on inputs and control functions.

**Claims And Evidence:**

This paper mainly proposes two claims:

1. The generalization analysis of controllable learning needs to address two key challenges, i.e., establishing the relationship between generalization bound and controllability and reducing the dependency of generalization bounds on the number of task targets.

2. Different manipulation methods based on the input and control function will lead to significant differences in the constant $A$ of the generalization bounds in Theorem 4.5 and 4.7.

The theoretical results in this paper provide clear and convincing evidence for these two claims.

**Essential References Not Discussed:**

This paper adequately discusses the related works.

**Experimental Designs Or Analyses:**

This is a purely theoretical paper. All the analyses in the Remarks in this paper are reasonable and valid. These analyses are well structured and well supported the claims of this paper.

**Methods And Evaluation Criteria:**

Although specific evaluation metrics are not available to measure the generalization performance of controllable learning methods, the two metrics analyzed in this paper are commonly used in practice.

**Other Comments Or Suggestions:**

I suggest providing proof sketches in the main paper to show the proof process and key theoretical techniques.

**Other Strengths And Weaknesses:**

Strengths:

1) This paper provides a theoretical foundation for understanding the generalization performance of controllable learning methods and for further analyzing a wider range of controllable learning methods. The proposed unified theoretical framework covers two typical controllable learning methods well, and the derived generalization bound has a weak dependency on the number of task targets, which is consistent with the practical situation, that is, it provides a theoretical guarantee that proper controllable learning methods can handle multiple task targets well.

2) This paper proposes a novel vector-contraction inequality specific to controllable learning. This inequality can induce a tight generalization bound with a logarithmic dependency on the number of task targets. It decouples the associations between multiple task targets, consistent with the potentially significant differences between task targets in practice and the losses corresponding to each task target in the weighted Hamming loss.

3) This paper provides generalization analysis for two typical controllable learning methods. The modular decomposition of theoretical results on specific controllable learning methods enhances readability and provides theoretical tools and preliminary results for further analysis in the future. It also provides an interface for subsequent expansion of modular theoretical results on more models. The theoretical techniques of deep models in this paper may be of independent interest.

4) This paper is well written and structured and easy to follow. The motivations and contributions are stated clearly. Each theoretical result is explained and analyzed in detail, making the theoretical results easy to understand. In addition, the generalization bounds of specific controllable learning methods are consistent with the selection of models in practice, which explains the empirical success of existing controllable learning methods.

Weaknesses:

The theoretical tools and intermediate steps involved in the proofs of theoretical results in the main paper are vague. Although detailed proofs are provided in the appendix, the lack of theoretical details in the main paper may make the theoretical contributions not better presented.

**Questions For Authors:**

Although specific evaluation metrics for the generalization performance of controllable learning are currently lacking, the question is whether the theoretical results obtained from the two losses involved in this paper are valid without loss of generality, that is, whether the theoretical results obtained from the relevant assumptions can be extended to potentially more evaluation metrics that will appear in the future.

**Relation To Broader Scientific Literature:**

Although existing controllable learning methods have demonstrated empirical success, theoretical research on controllable learning methods is completely unexplored. This paper lays the foundation for theoretically understanding the success of controllable learning methods from the perspective of generalization.

**Theoretical Claims:**

I have checked most of the theoretical proofs, and the proofs' correctness supports the claims made in the main paper.

---

> ### Author Rebuttal · Authors · 2025-03-31
>
> Thank you for your constructive comments and active interest in helping us improve the quality of the paper.
>
> The following are our responses to the Questions:
>
> **1. Response to the Weakness and Suggestion.**
>
> All proof sketches and detailed proofs of the theoretical results in this paper have been moved to the appendix due to the limitation of the paper length. We will add all proof sketches to the main body of the paper in the revised version to better demonstrate key theoretical techniques and improve the readability of the paper.
>
> **2. Response to the Question.**
>
> Although this paper only studies two loss functions, namely weighted Hamming loss and multi-target bipartite ranking loss, in fact our theoretical results are not limited to these two forms of loss. Note that our theoretical results only assume that the loss is Lipschitz continuous with respect to the $\ell_\infty$ norm. This assumption is relatively mild and easy to satisfy. It only requires that the derivative of the loss function is bounded, which can also be guaranteed to some extent by the boundedness of the loss function. Hence, for potential evaluation metrics in the future, it is only necessary to verify that they are Lipschitz continuous with respect to the $\ell_\infty$ norm, and the theoretical results here are also applicable to them, with only the difference of the Lipschitz constants $\mu$, as we showed in Proposition 4.3.

---

### Official Review · Reviewer_eMeP · 2025-03-11

**Overall Recommendation:** 4

**Summary:**

This work focuses on the generalization analysis of the controllable learning scenario. It establishes a unified theoretical framework and develops a novel vector-contraction inequality for controllable learning based on the Lipschitz continuity of loss functions.
The authors first formalize the function class of controllable learning framework, and derive a general Rademacher complexity upper bound for the loss function space based on the Lipschitz continuity and the Rademacher complexity of the controllable learning function class. Based on the general analysis, the authors give the Rademacher complexities and the generalization analyses for Hamming loss and ranking loss that is used in controllable learning.
For some specific learning methods, i.e., embedding-based methods and hypernetwork-based methods, the authors give the analyses of the empirical Rademacher complexities respectively. In addition, they also give the Rademacher complexity of the class of feedforward neural networks, graph neural networks and transformers.

**Claims And Evidence:**

This paper mainly considers the generalization ability of the controllable learning scenario. The authors use the empirical Rademacher complexity as the fundamental tool for generalization analysis, which is quite standard. I went through the proofs of the main theorems in this paper, and I think that the proofs are convincing and correct.

**Essential References Not Discussed:**

I don’t think that the authors miss any important related works. The authors cite most theoretical works of the generalization analysis of machine learning algorithms. Most of the cited works are state-of-the-art results in this area.

**Experimental Designs Or Analyses:**

This paper focuses on theoretical understanding, and hence no experiment is involved.

**Methods And Evaluation Criteria:**

The proposed theoretical results make sense for the problem. This paper studies the Hamming loss and ranking loss for controllable learning, which is widely adopted for this problem. The authors study the generalization ability of these two loss functions based on the Rademacher complexity, which is also a standard tool that is suitable for this problem.

**Other Comments Or Suggestions:**

The clarity of this paper would be further improved if the authors were to give some real-world examples of the controllable learning scenario.

**Other Strengths And Weaknesses:**

The detailed strengths of this work are as follows:
1.	This paper is well-written and easy to follow. The problems are clearly stated and the key contributions make sense.
2.	The originality of this paper is good. The authors combine the theoretical tool that is used to analyze the generalization ability of multi-label learning to the controllable learning scenario. This work gives the theoretical understanding of existing controllable learning algorithms, and hence have relatively significant impact on this area.

Despite the strengths above, this work has the following weaknesses:
1. The formulation of controllable learning in Section 3 is a little bit confusing. It could be better if some examples were provided to further explain the function class in Eqn. (2).

**Questions For Authors:**

What is the relationship between controllable learning and multi-label learning? It seems that, from Section 3, controllable learning is a special application of multi-label learning. Can the theoretical results generalize to general multi-label learning scenario?

**Relation To Broader Scientific Literature:**

I think that the key contributions of this paper are important. Since many algorithms for controllable learning is proposed, this paper first establishes a unified framework to analyze the generalization ability of controllable learning algorithms. It gives a formal definition of the function class of controllable learning, and apply the empirical Rademacher complexity to derive the generalization bound, which verifies the effectiveness of existing controllable learning algorithms theoretically.

**Theoretical Claims:**

I have roughly checked the correctness of two main theorems, and I think the proofs are correct. For Theorems 4.5 and 4.7, the proofs basically follow the standard Rademacher analysis for multi-label learning in [Zhang and Zhang, 2021], which is published in ICML 2024. I think the proofs of Theorems 4.5 and 4.7 are correct. For theorems in section 5, which give the Rademacher complexity upper bound for some specific models, e.g., feedforward neural networks, graph neural networks and transformers. The proofs basically follow the upper-bound of the operator norm of each component in deep learning and the Lipschitz continuity of activation functions, which is also quite standard, as in [Neyshabur et al., 2015; Bartlett et al., 2017; Trauger and Tewari, 2024].

---

> ### Author Rebuttal · Authors · 2025-03-31
>
> Thank you for your constructive comments and active interest in helping us improve the quality of the paper.
>
> The following are our responses to the Questions:
>
> **1. Response to Weakness.**
>
> We will add relevant explanations after Eqn. (2). For embedding-based controllable learning methods, the task requirement $\boldsymbol{z}$ can be editable user profiles, the control function $\psi$ often uses Transformers, and the nonlinear mapping $\zeta$ induced by classifier can use FNNs. For hypernetwork-based controllable learning methods, the task requirement can be a task indicator, the model corresponding to the control function is a hypernetwork, and the nonlinear mapping $\zeta$ induced by classifier can use GCN-based structures or Transformer-based models.
>
> **2. Response to Suggestion.**
>
> We provide two real-world examples of controllable learning scenarios as follows.
>
> 1. A news aggregator adjusts article rankings in real time using user-specified rules (e.g., exclude gaming content today) to promote diverse topics. The control function modifies inputs/parameters to enforce filters while keeping recommendations relevant.
>
> 2. A trading algorithm adapts portfolio strategies based on real-time goals (e.g., protect capital during market drops) to minimize losses and maximize risk-adjusted returns. The control function tunes parameters dynamically to balance objectives without retraining.
>
> These examples show how our framework enables systems to adapt inputs/parameters at test time to meet evolving task requirements (e.g., content preferences, market conditions) while ensuring generalization guarantees for targets like diversity and risk management.
>
> **3. Response to Question.**
>
> From the perspective of the model's output, the outputs of both controllable learning and multi-label learning can be expressed as vector-valued outputs. From the perspective of the model itself and the input, they are different. Controllable learning places more emphasis on task requirements, i.e., the learner can adaptively adjust to dynamically respond to the requirements of different task targets, while multi-label learning does not involve task requirements corresponding to different task targets, so controllable learning is not a special case of multi-label learning. However, when the control function $\psi$ in controllable learning is $\emptyset$, controllable learning will degenerate into a specific multi-label method, i.e., multi-label learning based on the label-specific representation learning strategy. At this time, the relevant theoretical results can be generalized to multi-label learning scenarios.

---

### Decision · Program_Chairs · 2025-05-01

**Decision:**

Accept (poster)

**Comment:**

This paper provides generalization bounds for "controllable learning", an extension of the multi-label learning setting where at in addition to the sample, the learner also receives a task requirement $z$ at test time. The proofs mostly rely on the construction of auxiliary datasets involving every (input, class) pair, similarly to [MC,MC1,MC2,MC3,ZZ].  Overall, the paper is **quite comprehensive**, tackling several architectures and loss functions, including transformers and even pairwise losses. The original version attempted to tackle GNNs, though in my opinion the results did not make sense.

Reviewers eMeP, Gt3V and Rc3E were very positive about the paper, all giving a score of 4. Reviewer 9TGj gave a score of 3 which was later updated to 4. Although reviewers eMeP, Gt3V and Rc3E claim to have thoroughly read the supplementary, they missed out on several obvious issues in the original submission: **firstly**, there was a minor error in the log factors in Lemma A.5, which was inherited from the previous literature. This has been **fixed** by relying on the correct version from [CRL].
**Secondly,** the original submission provided a "Rademacher complexity bound" of a node based graph neural network. Although the authors' answer was originally evasive, after further discussion, we agreed that the bound couldn't be used to prove any excess risk or generalization bound for any well defined learning setting, and the results were **removed (solved)**. **Lastly**, the results for the pairwise learning setting also involved an undefined learning setting, imprecise proofs and a final result which cannot be used directly to prove generalization bounds. This has since been more or less **resolved** thanks to continuous discussion with the authors.

**Novelty**:  While I don't share the enthusiasm of the reviewers as I personally feel the novelty is limited (compared to [MC,MC1,MC2,MC3]) by the high standards of ICML, the paper still provides original results in multiple scenarios and covers a relatively large scope.



**Correctness:** I thank the authors for their exceptional efforts fixing the paper during the discussion phase with me. They have **managed to address all of my concerns** in their revised version. Therefore, I am happy to recommend **acceptance**.
Whilst I believe the three major correctness issues have been solved and the proofs are now correct at a high level, I *strongly advise the authors to thoroughly double check the paper for any remaining inaccuracies* before the camera ready deadline. From the revisions performed during the revision phase, I believe the authors are capable of it.


----------------------


[MC] Y. Lei, Ü. Dogan, D.-X. Zhou, and M. Kloft, "Data-Dependent Generalization Bounds for Multi-Class Classification", IEEE Transactions on Information Theory 2019. See arXiv https://arxiv.org/pdf/1706.09814

[MC1] L. Wu, A. Ledent, Y. Lei, and M. Kloft, "Fine-grained Generalization Analysis of Vector-Valued Learning", AAAI 2021.

[MC2] W. Mustafa, Y. Lei, A. Ledent, and M. Kloft, "Fine-grained Generalization Analysis of Structured Output Prediction", IJCAI 2021.

[MC3] Hieu et al. Generalization Analysis for Deep Contrastive Representation Learning, AAAI 2025

[ZZ] Zhang and Zhang, Generalization analysis for multi-label learning. ICML 2024.

[CRL] Yunwen Lei, Tianbao Yang, Yiming Ying, Ding-Xuan Zhou, “Generalization Analysis for Contrastive Representation Learning”, ICML 2023.